# Protein biogenesis machinery is a driver of replicative aging in yeast

Georges E Janssens[1†], Anne C Meinema[2†‡], Javier González[3§], Justina C Wolters[4], Alexander Schmidt[5], Victor Guryev[1], Rainer Bischoff[4], Ernst C Wit[3], Liesbeth M Veenhoff[1]*, Matthias Heinemann[2]*

[1]European Research Institute for the Biology of Ageing, University Medical Center Groningen, University of Groningen, Groningen, The Netherlands; [2]Molecular Systems Biology, Groningen Biomolecular Sciences and Biotechnology Institute, University of Groningen, Groningen, The Netherlands; [3]Probability and Statistics, Johann Bernoulli Institute of Mathematics and Computer Science, University of Groningen, Groningen, The Netherlands; [4]Analytical Biochemistry, Groningen Research Institute of Pharmacy, University of Groningen, Groningen, The Netherlands; [5]Biozentrum, University of Basel, Basel, Switzerland

*For correspondence: l.m. veenhoff@rug.nl (LMV); m. heinemann@rug.nl (MH)

[†]These authors contributed equally to this work

Present address: [‡]Institute of Biochemistry, Zürich, Switzerland; [§]Sheffield Institute for Translational Neuroscience, Department of Computer Science and Department of Chemical and Biological Engineering, University of Sheffield, Sheffield, United Kingdom

Competing interests: The authors declare that no competing interests exist.

**Abstract** An integrated account of the molecular changes occurring during the process of cellular aging is crucial towards understanding the underlying mechanisms. Here, using novel culturing and computational methods as well as latest analytical techniques, we mapped the proteome and transcriptome during the replicative lifespan of budding yeast. With age, we found primarily proteins involved in protein biogenesis to increase relative to their transcript levels. Exploiting the dynamic nature of our data, we reconstructed high-level directional networks, where we found the same protein biogenesis-related genes to have the strongest ability to predict the behavior of other genes in the system. We identified metabolic shifts and the loss of stoichiometry in protein complexes as being consequences of aging. We propose a model whereby the uncoupling of protein levels of biogenesis-related genes from their transcript levels is causal for the changes occurring in aging yeast. Our model explains why targeting protein synthesis, or repairing the downstream consequences, can serve as interventions in aging.

## Introduction

Aging, the gradual decrease in function occurring at the molecular, cellular, and organismal level, is a main risk factor for cardiovascular disease, neurodegeneration, and cancer (*Niccoli and Partridge, 2012*). Understanding its driving force is the required step towards enabling interventions that might delay age-related disorders (*de Magalhães et al., 2012*). While this remains an unsolved problem in biology (*Medawar, 1952*; *Mccormick and Kennedy, 2012*), significant advances in the field have shown the process of aging to be malleable at both the genetic and environmental levels, indicating that it is possible for its causal elements to be dissected. The rate of aging, however, is influenced by diverse factors, including protein translation, protein quality control, mitochondrial dysfunction, and metabolism (*Kennedy and Kaeberlein, 2009*; *Webb and Brunet, 2014*; *Lagouge and Larsson, 2013*; *Barzilai et al., 2012*). The multitude of factors involved indicates that aging is a complex and multifactorial process, where ultimately an integrated and systems-level approach might be necessary to untangle the causal forces.

Important insights into the complex process of aging originate from research on the unicellular eukaryote *Saccharomyces cerevisiae,* which can produce 20–30 daughter cells before its death (*Mortimer and Johnston, 1959*, and see *Wasko and Kaeberlein, 2014*; *Denoth Lippuner et al.,*

**eLife digest** Aging is a complex process, and so many scientists use baker's yeast as a simpler model to understand it. Although many genes that influence aging have been found, all the generated knowledge is still rather fragmented. It also remains difficult to disentangle cause and consequence. That is to say, sometimes a gene that looks like it might cause aging could simply be a gene that responds to an age related phenomenon. To unravel this puzzle of cause and effect, it is necessary to first get an idea on a system level of everything that changes as an organism ages.

Now, Janssens, Meinema et al. have managed to map many of the molecular changes that occur as baker's yeast ages; this is something that has yet to be achieved for any other organism. The work first involved developing a new way of growing baker's yeast to keep and generate large cohorts of aging yeast cells in a constant environment. It also required the use of a mathematical 'un-mixing' tool to separate the data obtained from the aging cohort from the data from the young offspring that the yeast produce while they age.

Janssens, Meinema et al. measured both the majority of the transcriptome and much of the proteome of baker's yeast throughout its reproductive lifespan. The "transcriptome" refers to the collection of RNA molecules in the cell, which are produced whenever a gene is expressed. The "proteome" refers to all the proteins in the cell, which are translated from the RNA transcripts by the cell's so-called "translational machinery". These experiments revealed that this yeast's proteome reflects its transcriptome less and less as it ages. In particular, this 'uncoupling' of the proteome from the transcriptome was seen most strongly for the proteins related to the cell's translational machinery; these proteins accumulated with age relative to their transcripts.

Janssens, Meinema et al. then conducted a computational network-based analysis of the data. This indicated that the uncoupling is the driving force behind the aging process. Many of the other molecular changes that occur with aging were predicted to be consequences of this uncoupling.

These findings give a framework for many observations in the existing literature. However, it remains unclear why proteins related to translational machinery are overrepresented in aging yeast in the first place. This question should be explored in future work.

*2014* for recent reviews). Significant contributions towards global mapping of the aging process have been demonstrated through transcriptome studies (*Egilmez et al., 1989*; *Lin et al., 2001*; *Lesur and Campbell, 2004*; *Koc et al., 2004*; *Yiu et al., 2008*) and genome-wide single-gene deletion lifespan measurements (reviewed in *Mccormick and Kennedy, 2012*). However, a major task remains to comprehensively describe the molecular changes that accompany the aging process. As the exponential increase in daughter cells represents a major challenge in terms of generating sufficient numbers of aged cells, to date no comprehensive description of the changes on both the proteome and transcriptome level has been provided. Assuming that the molecular changes occurring along the replicative lifespan of yeast are, in part, responsible for its decreased viability that occurs over time, we reason that revealing the dynamic and interdependent changes that accompany this process would allow us to distinguish cause from consequence in aging.

Here, we developed a novel column-based cultivation method that allowed us to generate large numbers of advanced-age cells in a constant environment. Applying next-generation RNA sequencing and shotgun proteomics, we mapped the molecular phenotypes of aging yeast cells at 12 time points, well into advanced age where the majority of cells had died due to aging. Analysis of these dynamic and comprehensive datasets allowed us to identify a general uncoupling of protein levels from their corresponding messenger RNA (mRNA) levels. This uncoupling was most apparent in protein biogenesis-related proteins, which we found over-represented relative to their transcripts. Using computational network-based inference methods, we found that changes in these genes had the strongest ability to predict the behavior of other genes, thereby suggesting their causal role in replicatively aging yeast. On the basis of these analyses, we provide a systems-level model of aging unifying and integrating diverse observations made within the field.

## Results

### Novel culture and computational methods to determine aged cell phenotypes

To obtain aged yeast cells, we bound streptavidin-conjugated iron beads to biotinylated cells (adapted from *Smeal et al., 1996*) from an exponentially growing culture. This starting cohort of mother cells was put into a column containing stainless steel mesh that was positioned within a magnetic field (*Figure 1A*, *Figure 1—figure supplement 1*). The daughter cells do not inherit the iron beads, as the yeast cell wall remains with the mother during mitosis (*Smeal et al., 1996*). By running a constant flow of medium through the column, we washed away the majority of emerging daughter cells. The flowing medium also provided fresh nutrients and oxygen and ensured constant culture conditions, as confirmed for pH, glucose, and oxygen levels (*Figure 1—figure supplement 2A–C*). By maintaining multiple columns simultaneously, we could harvest cells from the same starting cohort at different time points and thus at different replicative ages (*Figure 1—figure supplement 2D*). Because we could retain up to $10^9$ mother cells per column (*Figure 1—figure supplement 3*), we could produce sufficient numbers of aged cells for performing parallel proteome and transcriptome analyses. Computer simulations showed that the age distribution broadened over time (*Figure 1—figure supplement 4A,B*). The broadened age distribution results in a lower resolution making detecting the actual changes occurring at later time points more difficult, and we therefore harvested cells at exponentially increasing time intervals to maximize the differences between time points at later ages.

To assess whether our column-based cultivation method generated correctly aged cells in a reproducible manner, we developed flow cytometric assays to determine the typical phenotypes of aging cells. Avidin-fluorescein isothiocyanate (AvF) binding to the biotin-labeled cells distinguished the starting cohort of mother cells from daughter cells (*Figure 1—figure supplement 5A*). Dead cells were identified using propidium iodide (PI), which fluoresces upon intercalating with the DNA of membrane-permeable dead cells (*Figure 1—figure supplement 5A*). These two assays were used to determine the fractions of daughters, mothers, and dead cells in a population (*Figure 1—figure supplement 5B*). From this data, we derived the viability of the mother cells over time, which we found to be in excellent agreement with the lifespan curve of yeast as observed in a microfluidic device (*Huberts et al., 2014*) (*Figure 1B*). Using the forward scatter of the flow cytometer as a rough proxy for cell size, we could qualitatively observe the cell size increase of live mothers that is known to occur in aging mother cells (*Egilmez et al., 1990*) (*Figure 1C*). Similarly, using fluorophore-conjugated wheat-germ agglutinin, which labels bud scars that appear after every division (*Powell et al., 2003*), we observed an increase of bud scar staining on mother cells in the column, as also visualized by confocal microscopy (*Figure 1D*, *Figure 1—figure supplement 2D*). These analyses confirmed known changes that characterize aging yeast: increased cell size and bud scars, and decreased population viability (*Figure 1B–C*).

Next, we developed a combined experimental and mathematical method to determine the molecular phenotype of aging mother cells without contributions from daughter or dead cells. The approach exploits the fact that a system of linear equations can be solved when the number of unknowns equals the number of independent equations. Specifically, while we could determine the number of mothers, daughters, and dead cells in a sample using flow cytometry, the contribution of each type of cells to the measured abundance of a particular protein or transcript was unknown. Therefore, by measuring protein and transcript abundances in three mixed samples with various proportions of mothers, daughters, and dead cells, we could mathematically un-mix the abundances. This resulted in un-mixed data for the aging mother cells. Experiments using samples containing mixed cell populations with known molecular phenotypes validated this mathematical un-mixing method for the RNA sequencing (RNAseq) transcriptome, targeted (selected reaction monitoring) proteome, and global (shotgun) proteome data with a <16% average error (*Figure 2—figure supplement 1 and 2*; *Supplementary file 1*).

To use this data un-mixing approach, we harvested three mixed samples for each time point (*Figure 2A*, *Figure 2—figure supplement 3*). One sample was collected from the column effluent (Mix 3, mainly daughter cells). Harvesting all cells from the column and applying a further enrichment step on a larger magnet produced the two other samples: one sample contained mainly aged mother cells (Mix 2, 80–99% mothers), while the other contained an intermediate composition

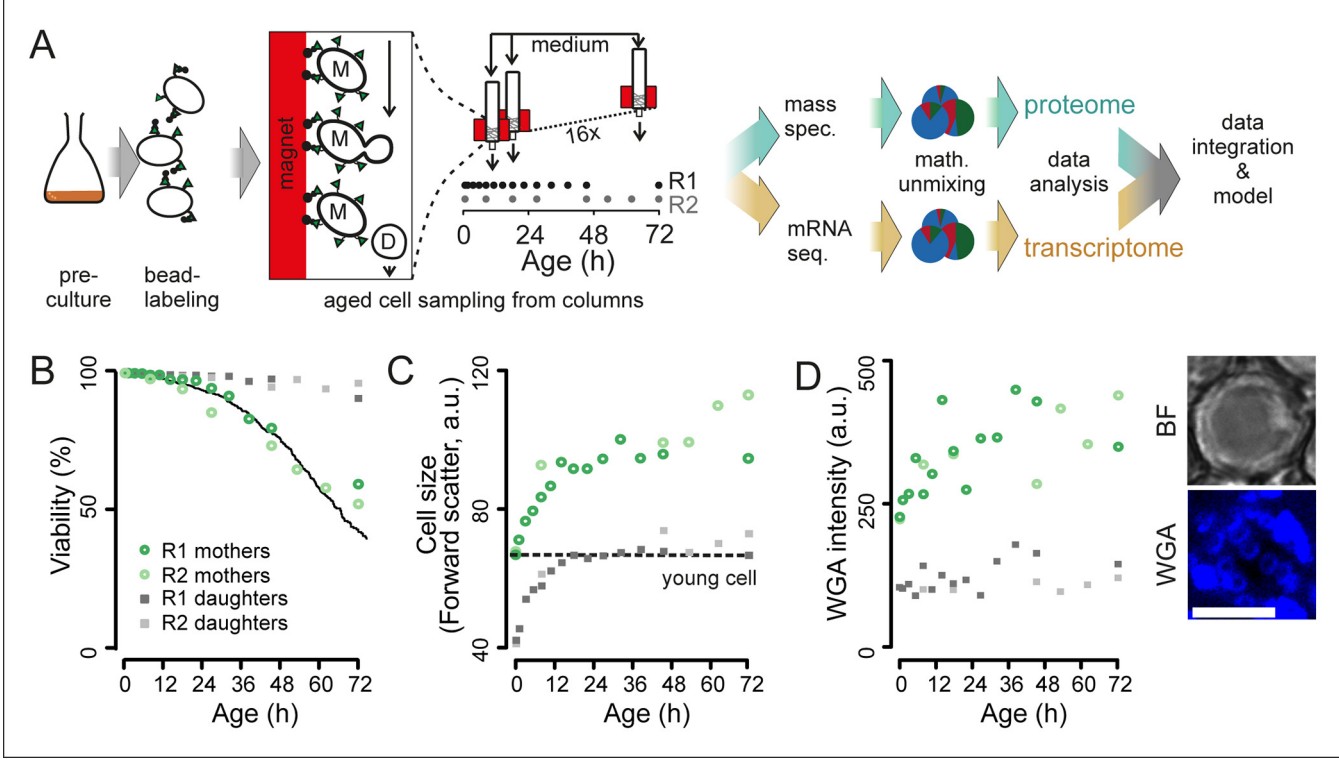

**Figure 1.** Experimental design for analysis of molecular changes during the replicative lifespan of yeast and its validation. (**A**) Schematic overview of the column-based cultivation and data analysis pipeline with 16 parallel columns, where (zoom in) mother cells (M) containing streptavidin-bound (green triangles) iron beads (black circles) were captured on the magnetized column and aged under constant environmental conditions, while the daughter cells (D) were flushed away. Samples are collected in two replicate campaigns (R1, R2) at indicated time points in the lifespan. (**B**) Flow cytometry-based assessment of viability of mother (Avidin-fluorescein isothiocyanate positive [AvF]) and daughter (AvF negative) cells in R1 and R2, calculated for each time point comparing viable (propidium iodide [PI] negative) versus inviable (PI positive) cells in harvested samples Mix 1–3 (see *figure 2A* for explanation of Mix 1–3). The solid black line represents cell viability in time measured for the same strain in the same media using a microfluidic device (*Lee et al., 2012*; data from *Huberts et al., 2014*, was obtained from the authors). (**C**) Cell size is qualitatively assessed with median forward scatter of live mothers (AvF positive, PI negative) vs live daughters (AvF and PI negative). Dashed line represents the median forward scatter of young cells that have reached the fully-grown cell size to start their first division. (**D**) Aging was qualitatively assessed throughout the experiment by observing an increase in median WGA intensity over time in a population of primarily mothers (Mix 2) compared to a sample composed primarily of daughters flushed out of the column (Mix 3). Inset: bright field (BF) and fluorescence microscopy image of cell stained with AlexaFluor 633 conjugated wheat germ agglutinin (WGA), which selectively binds chitin in bud scars. Scale bar 5 μm.

The following source data and figure supplements are available for figure 1:

**Source data 1.** Table S1: Materials used for construction of novel column-based cultivation method.

**Figure supplement 1.** Setup of the aging columns.

**Figure supplement 2.** Cellular aging under constant conditions.

**Figure supplement 3.** Cell counts per time point.

**Figure supplement 4.** Simulated yeast aging population dynamics.

**Figure supplement 5.** Characterization of mixed-cell samples.

compared to Mixes 2 and 3 (wash fraction, Mix 1). In each of these mixed-cell samples, we determined the fraction of mothers, daughters, and dead cells and generated the mixed-population proteomes and transcriptomes. Then, we mathematically un-mixed the proteomes and transcriptomes to obtain the molecular phenotype of aging mother cells. The data was corrected for sampling

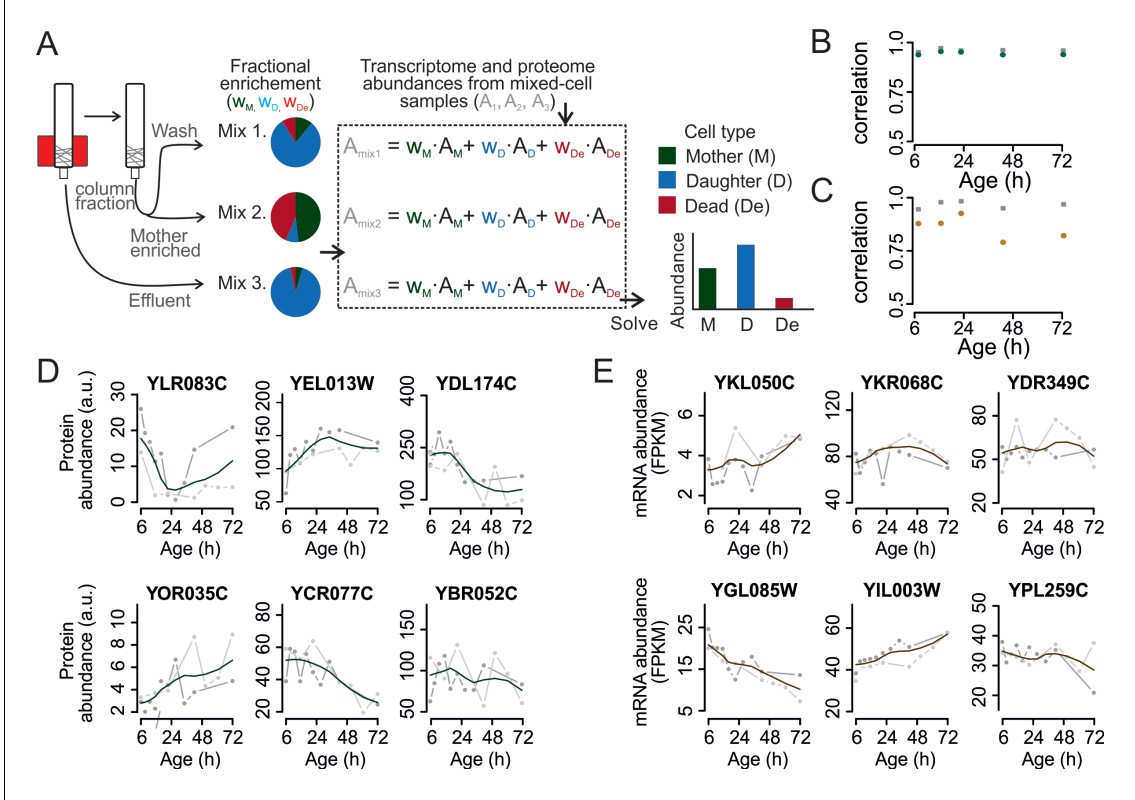

**Figure 2.** Mathematical un-mixing of proteomes and transcriptomes in mixed-cell populations. For each time point in the aging experiment, three samples (mixed-cell samples 1,2,3; originating from different harvesting steps) composed of different fractions of Mother (M, green), Daughter (D, blue) and Dead cells (De, red) were harvested and analyzed. On the basis of the compositions of the mixed-cell samples ($w_{M, D, De}$) and the determined proteome or transcriptome data of the mixed-cell samples ($A_{mix1,2,3}$), with the mathematical un-mixing, we obtained un-mixed data ($A_{M, D, De}$) over the time course of 72 hr from two replicates. See *Figure 1—figure supplement 5* for details about determining the composition of the mixed-cell samples and *Figure 2—figure supplement 3* for the un-mixing method. Data from proteome (**B**) and transcriptome (**C**) replicates highly correlated (Spearman correlation >0.85) for M (circles) and D cells (squares), indicating high reproducibility of the experimental and data processing pipelines. (**D, E**) Levels of random chosen proteins (**D**) and transcripts (**E**) from both replicate measurements (gray) and the fit (solid line) are indicated for un-mixed mother data. Raw abundance is a measure of mass spectrometry (MS) peak intensities (proteome) or fragments per kb of transcript per million mapped (FPKM) reads (transcriptome).

The following source data and figure supplements are available for figure 2:

**Source data 1.** Table S2: The shotgun proteome data processing.

**Source data 2.** Table S3: The transcriptome data processing.

**Source data 3.** Table S4: The final shotgun proteome data.

**Source data 4.** Table S5: The final transcriptome data.

**Figure supplement 1.** Validation of the mathematical un-mixing procedure.

**Figure supplement 2.** Validation of the mathematical un-mixing procedure, shotgun proteome and RNA sequencing.

**Figure supplement 3.** Generation and composition of the mixed-cell samples.

**Figure supplement 4.** Validation of the bead effect correction.

**Figure supplement 5.** Overview of the experimental pipeline.

*Figure 2. Continued*

**Figure supplement 6.** Selection of genes with highest similarity between replicates.

artefacts related to bead labeling and cell harvesting (*Figure 2—figure supplement 4*; supplemental notes 2 and 3 in *Supplementary file 1*). Together, through this approach, we obtained pure data for aging mother cells and daughter cells.

In two experimental series with overlapping time points, we generated 61 samples for both the proteome and the transcriptome as required for un-mixing. After data processing, we obtained high quality data at 12 unique time points during the lifespan of replicatively aging yeast (*Figure 2—figure supplement 5*). We found the replicates to be in excellent agreement (Spearman correlations >0.85) (*Figure 2B,C*). A unified dataset was generated for both the proteome and the transcriptome by fitting the replicate datasets with a polynomial regression (*Figure 2D,E*), only retaining highly reproducible expression profiles (~85% of genes, *Figure 2—figure supplement 6*), and resampling the fit at the actual time points of the experiment. This yielded profiles for 1494 proteins and 4904 transcripts from aging mother cells. The raw data (*Janssens et al., 2015a*; *Janssens et al., 2015b*) and the data for each processing step are provided in the supplementary Tables S2 and S3 (*Figure 2—source data 1 and 2*). The final datasets for aging mother cells are presented in Table S4 (proteome) and Table S5 (transcriptome) (*Figure 2—source data 3 and 4*).

## Biogenesis proteins increase relative to transcript levels during aging

Correlation analyses between the proteomes of young cells and the proteomes of aging mother cells confirmed the expected divergence of the aging cell away from the youthful state (*Figure 3A*, *Figure 3—figure supplement 1*). Daughters from later time points showed a partially aged signature (*Figure 3—figure supplement 2*), consistent with the notion that rejuvenation of daughter cells is incomplete later in a mother's life (*Kennedy et al., 1994*). Furthermore, we found agreement between specific proteome changes detected by us and observations present in literature, including changes related to glycolysis, gluconeogenesis (*Lin et al., 2001*), increased expression levels in energy reserve pathway proteins (*Levy et al., 2012*), increases in stress response protein levels (*Erjavec et al., 2007*; *Crane et al., 2014*), and mitochondrial changes (*Hughes and Gottschling, 2012*) (*Figure 3B*, *Figure 3—figure supplement 3*). Also, we confirmed that changes detected in our population-level study similarly occurred at the single-cell level, which excluded the possibility that our observed changes may reflect a gradual enrichment of a long lived subpopulation. Specifically, we see the levels of the stress-related chaperone Hsp104 and the translation elongation factor Tef1 to increase with age (*Figure 3—figure supplement 4*), similar to what was shown using a microfluidic platform tracking single cells (*Zhang et al., 2012*). Also, other single protein changes reported to occur in literature match well (*Koc et al., 2004*; *Lee et al., 2012*; *Hughes and Gottschling, 2012*; *Zhang et al., 2012*; *Lord et al., 2015*; *Denoth-Lippuner et al., 2014*; *Eldakak et al., 2010*; *Sun et al., 1994*) (*Figure 3—figure supplement 4*). Together, these observations confirm the validity of our novel experimental design.

To obtain further insights into the global changes in protein expression in mother cells, we plotted our dynamic data as heat map expression profiles. We found that changes started at young age, were gradual, and mostly occurred in one direction (i.e. up, down) (*Figure 4A,B*). Specifically, we found that 64% (184/288 total changes) of the proteins that showed a twofold change by the end of the yeast lifespan also showed a significant change in the same direction at an earlier time point (*Figure 3B*). These findings suggest that aging is a gradual process occurring from early on.

We next investigated whether these changes in the proteome data matched transcriptional changes. Interestingly, the RNAseq data showed similar gradual and unidirectional changes occurring from the beginning on (*Figure 4—figure supplement 1A,2,3*). To compare the changes between the proteome and transcriptome, we determined the non-parametric Spearman rank correlation, and found a starting correlation of 0.75, a value in agreement with other single-study comparisons between yeast proteomes and transcriptomes (*Csárdi et al., 2015*). When comparing this correlation in time, however, we found that it declined steadily with age, down to a correlation of 0.70 (*Figure 5A*). This decreasing trend was observed regardless of the statistical method used (*Figure 5—figure supplement 1*). Furthermore, this trend is also not an experimental artefact, since

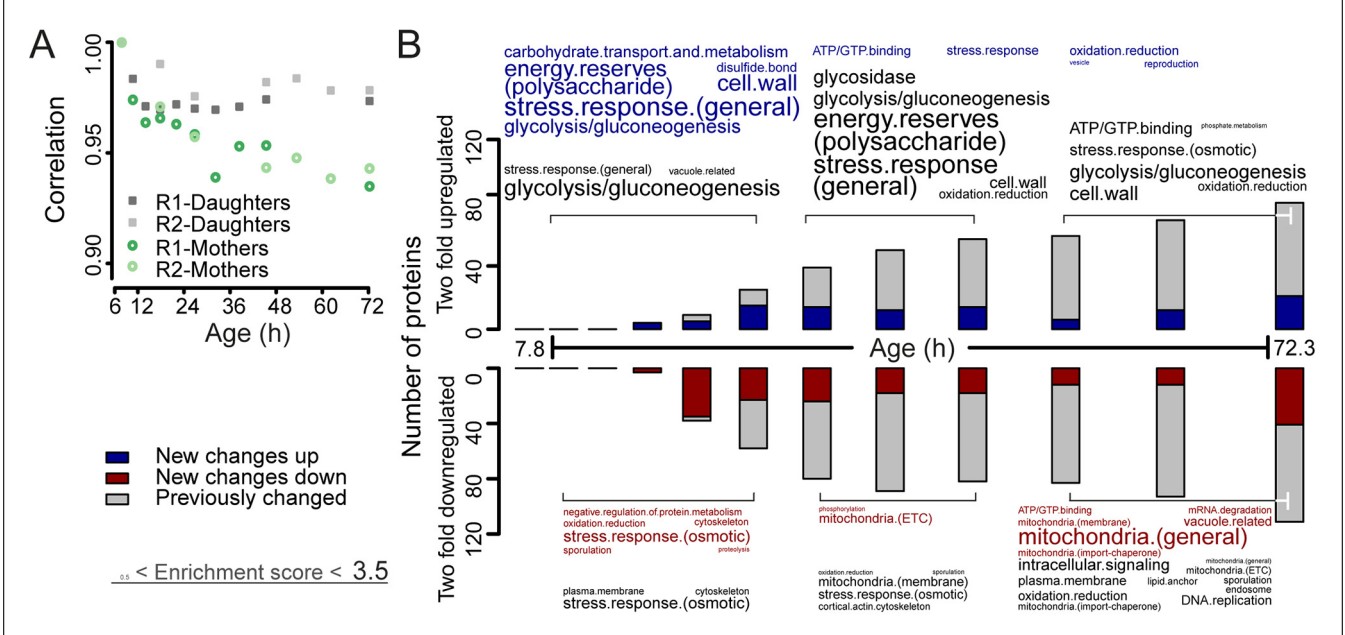

**Figure 3.** The aging proteome. (**A**) The Spearman correlation at progressive time points compared with the young reference sample for the mother and daughter proteome shows a divergence away from a youthful state for the mother. (**B**) The numbers of proteins changing by at least twofold from the reference (young) sample per time point. Blue and red bars and text represent changes that had not occurred previously, either up- or down-regulated, respectively. Gray bars and text are changes that already occurred at a previous time point. Gene functional enrichments per grouped time points were derived from Gene Ontologies (GO) and are scaled with significance of enrichment obtained by database for annotation, visualization and integrated discovery (DAVID) bioinformatics resource version 6.7 (scaling of text: DAVID enrichment score see Materials and methods and Table S6 (*Figure 3—source data 1*).

The following source data and figure supplements are available for figure 3:

**Source data 1.** Table S6: Full lists of GO-term enrichment scores for all enrichment analyses.

**Figure supplement 1.** The aging transcriptome diverges minimally from a young profile.

**Figure supplement 2.** Changes in mother-age dependent daughter profiles.

**Figure supplement 3.** Profiles that contribute to the enrichments of proteins changing more than twofold.

**Figure supplement 4.** Single protein profiles matching literature.

samples originating from all time points were treated identically, and both proteome and transcriptome datasets originated from the same biological samples. The decrease in correlation between the proteome and transcriptome means that they do not change synchronously. Indeed, during aging, we found different Gene Ontology (GO) terms to describe the changes in the proteins and transcripts that show a larger than two-fold change during aging (*Figure 3B* vs. *Figure 4—figure supplement 2A*). These results indicate that, over time, protein levels were increasingly uncoupled from their transcript levels.

To identify the most uncoupled cellular processes, we plotted the fold-changes of transcript and protein expression in old and young cells on a gene product co-expression map (*Figure 5A*). The transcript and protein levels of genes in quadrants 1 (Q1) and 3 (Q3) were 'coupled', meaning that the changes in protein levels followed the changes in transcript levels. Q1 and Q3 were enriched in gene products related to sterol biosynthesis and cytoskeletal and cell wall processes, possibly related to cell growth. In contrast, the expression of gene products in quadrants 2 (Q2) and 4 (Q4) were 'uncoupled', meaning that the changes in protein levels did not follow the changes in transcript levels. In Q2, the proteins were over-represented relative to their transcripts, that is, there were

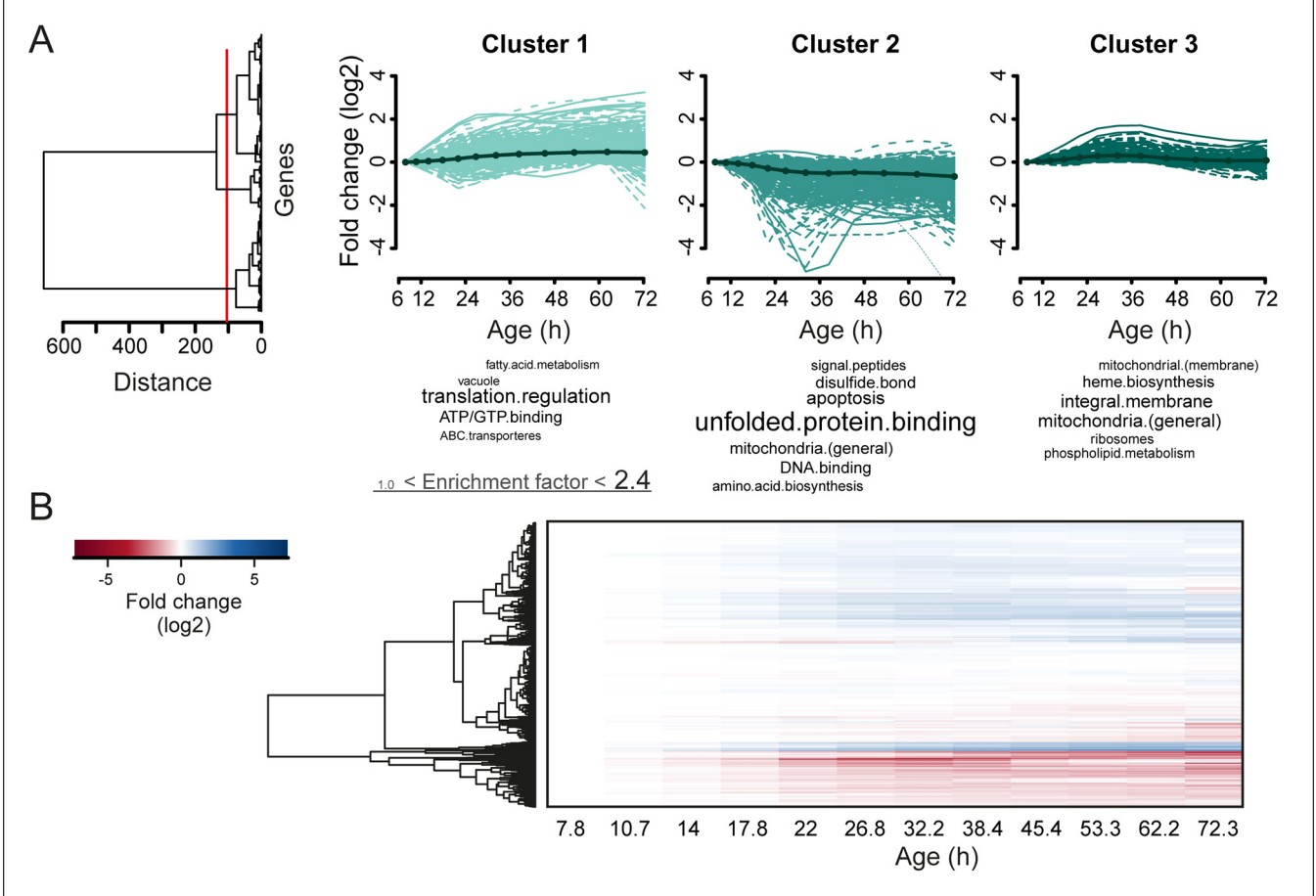

**Figure 4.** Protein profiles in aging yeast. (**A**) Expression profiles for the proteome were clustered using the Ward clustering algorithm and plotted in a dendrogram. Visualization of the most prominent (red line in dendrogram) protein fold change profiles (log$_2$ scale) occurring with age, showing up-regulated (cluster 1), down-regulated (cluster 2) and mainly flat (cluster 3) profiles. Gene functional enrichments per grouped time points were summarized into representative terms as in *Figure 3B*. (**B**) Unidirectional changes occurring with aging are illustrated with a heat map of the fold changes (log$_2$ scale) of proteins in the aging mother compared to the young reference sample.

The following figure supplements are available for Figure 4:

**Figure supplement 1.** Comparison of aging proteomes and transcriptomes.

**Figure supplement 2.** Analysis of twofold changes per time point in the aging transcriptome.

**Figure supplement 3.** Analysis of aging changes clustered by expression profile.

more proteins per transcript in older cells than in younger cells. Of all analyzed transcript–protein pairs, 38.4% were located in Q2, suggesting a global tendency towards relative protein overabundance with aging (*Figure 5*). In line with this global protein overabundance, Q4 contained fewer genes and less GO-term enrichments. Strikingly, Q2 was strongly enriched in 'translation regulation' gene products (i.e. ribosome and protein biogenesis machinery) (*Figure 5B*), and the extent of their overabundance progressively increased as the cells aged (*Figure 5—figure supplement 2,3*).

## Network inference identifies protein biogenesis-related genes as causal in yeast aging

Next we asked whether this increased level of biogenesis-related proteins, uncoupled from transcriptional regulation, was causal for downstream effects during replicative aging in yeast. Identifying causality on a systems-wide level is difficult, and the key challenge is to separate cause from

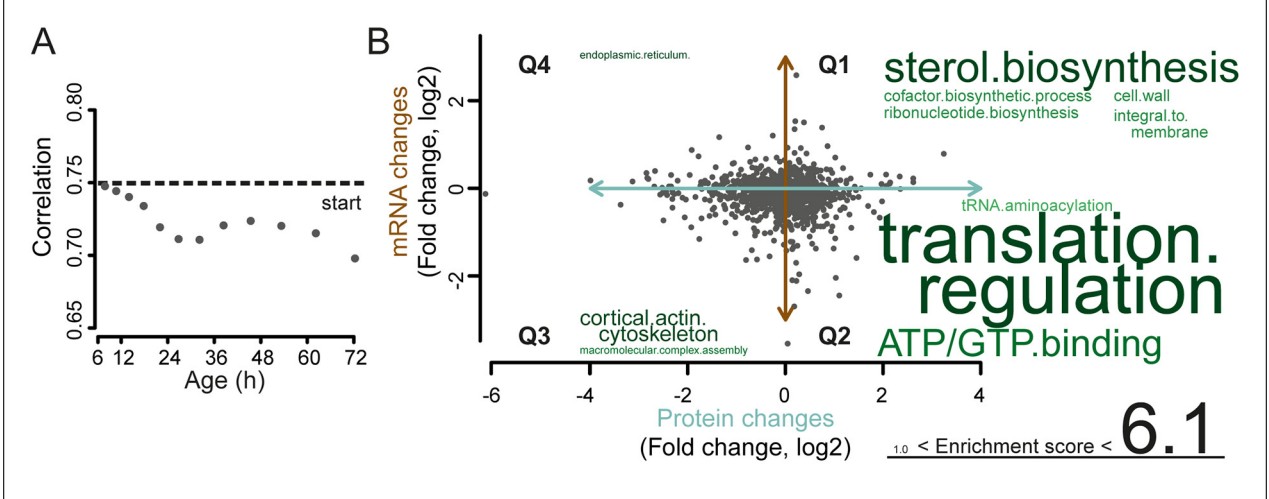

**Figure 5.** A post-transcriptional overrepresentation in protein biogenesis with aging. (**A**) A progressive uncoupling of the proteome from the transcriptome in time is apparent from the decreasing Spearman correlation between the two. (**B**) Co-expression map showing fold changes (log$_2$) of 72 hr aged samples compared to the young reference, plotting the proteome versus the transcriptome. Quadrants 1 and 3 (Q1 and Q3) represent changes where the protein changes match their transcript changes (coupled), while quadrants 2 (Q2) and 4 (Q4) reflect opposite changes (uncoupled). Summarizing terms per quadrant are derived from Gene Ontologies (GO) as in *Figure 3B* (scaling of text: DAVID enrichment score).

The following figure supplements are available for Figure 5:

**Figure supplement 1.** Correlation of proteome versus transcriptome using alternative statistical methods for comparison.

**Figure supplement 2.** Co-expression map showing fold changes of 10.7, 22, 45.4 and 72.3 hr compared to the young reference, highlighting gene products contributing to gene enrichments.

**Figure supplement 3.** Change in posttranscriptional protein overabundance with aging.

downstream effects. However, our dynamically resolved, comprehensive data offered the possibility to suggest causal relationships.

To elucidate the causal order of changes during aging, we reconstructed a high-level directional network revealing the interdependences of changes in transcript expression (*Figure 6*, *Figure 6— figure supplement 1A*). Therefore, we defined each transcript's expression profile as a network node, and an edge between each pair of nodes as a partial correlation between the nodes' expression profiles (*Figure 6—figure supplement 1B and C*). Next, we determined the directionality of the edges, indicated by arrows. We defined directionality to represent the ability of a transcript's profile to predict the profile of another transcript. Concretely, when looking at two connected nodes, the node that could be explained by the connected node was considered as the responsive node, while the predicting node was considered to be the causal node (*Opgen-Rhein and Strimmer, 2007*) (*Figure 6—figure supplement 1D and E*). This relation defined the directionality of the edge. Any transcript that had no predictive ability and could not be predicted by any other transcript was removed from the network analysis. Following this, the nodes were clustered by maximizing the global modularity of the network (*Csardi and Nepusz, 2006*) (*Figure 6A*). Finally, the clusters were ranked based on the ratio of causal (outward arrows) to responsive nodes (inward arrows) per cluster to determine the higher-level causal relations existing between clusters. A sensitivity analysis was performed to determine the optimal sparsity of the network and the cut-off for the partial correlation among transcript profiles, through which we established that the network was a robust representation of the datasets (supplemental note 4 in *Supplementary file 1*, Table S7, *Figure 6–source data 1*). These steps produced a high-level directional network, in which the ranking of the clusters with respective GO enrichments revealed causal relations during aging (*Figure 6B*).

This high-level directional network of the transcriptome data showed that the very first causal-ranked cluster in the network that we detected was highly enriched for gene products associated

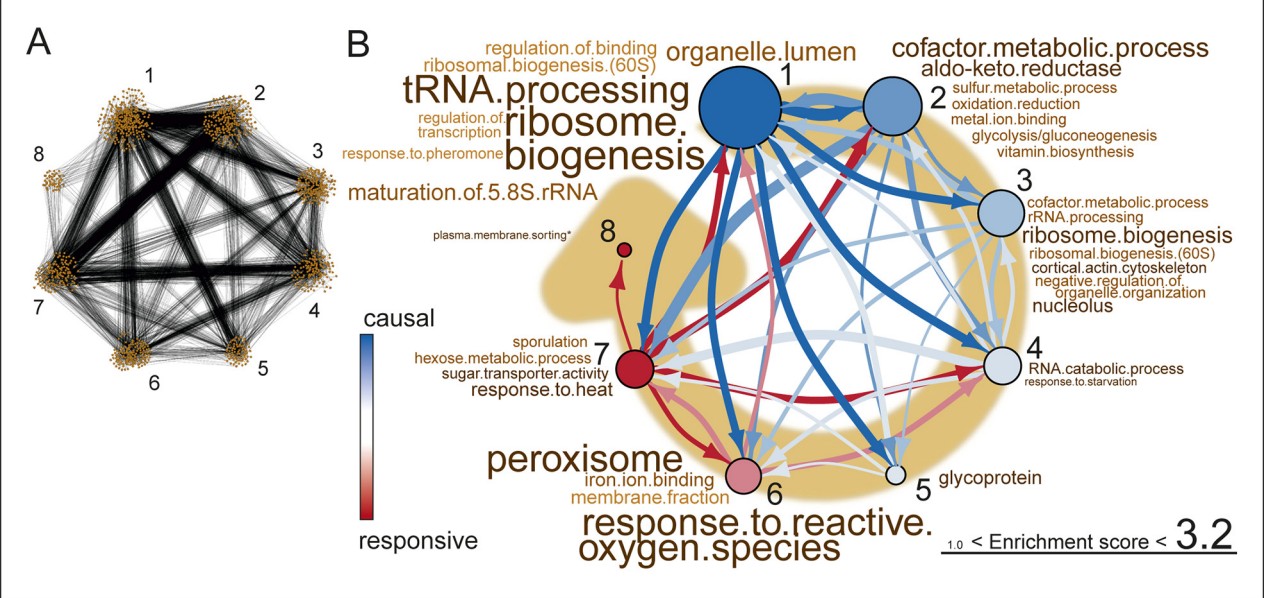

**Figure 6.** Network inference identifies protein biogenesis-related genes as causal force during aging. (A) The directed and clustered transcriptome network consists of 3631 edges, connecting 1241 nodes in 8 clusters (see *Figure 6—figure supplement 1* and supplemental note 4, in *Supplementary file 1*, for further details). Only actual relations are depicted, the causal direction between two nodes is indicated with an arrow, where the arrowhead points to the responsive node. (B) Clusters ranked from more causal to more responsive in the causality network (from blue to red for clusters 1 through 8). The degree of causality is determined by the ratio of the outgoing over incoming connections per cluster (from A). The blue to red arrows indicate the sum of outgoing arrows between two clusters, where arrow thickness is logarithmically scaled to the number of arrows (from A), that is, the summed predictive power of one cluster over the other. Terms per cluster are derived from Gene Ontologies (GO) as in *Figure 3B* (scaling of text: database for annotation, visualization and integrated discovery [DAVID] enrichment score).

The following source data and figure supplements are available for figure 6:

**Source data 1.** Table S7: The direction matrices and the sensitivity analyses for the proteomic and transcriptomic high-level directional networks.

**Figure supplement 1.** The transcriptome network.

**Figure supplement 2.** Network cluster gene enrichments in the co-expression map.

with protein biogenesis (i.e. ribosome biogenesis and transfer RNA [tRNA] processing; *Figure 6B*). These are the same biological processes that had uncoupled transcript and protein levels (*Figure 5B*); indeed, genes from this causal cluster were enriched in Q2 of the co-expression map, which showed uncoupled expression (*Figure 6—figure supplement 2A and B*). These analyses suggest that the uncoupling of protein and transcript levels for 'biogenesis'-related genes has a central role in the aging process, and may affect the transcript and protein abundances of other genes, as elaborated upon in the discussion.

## Consequences for other cellular processes

The overabundance of proteins relative to transcripts must have consequences for cellular functioning. Protein overproduction could increase cell size, one of the first hallmarks described in yeast aging (*Egilmez et al., 1990*). Increased cell size could reduce glucose influx rates per cell volume and induce metabolic changes, for example, at low rates of glucose influx, cells switch to respiration (*Huberts et al., 2012*). Indeed, in our transcript-based network analysis (*Figure 6B*) as well as in our proteome dataset (*Figure 3B*), we found that metabolic signatures related to starvation and oxidative stress were consequences of aging.

Furthermore, we hypothesized that if protein levels become globally uncoupled from their transcript levels during aging (*Figure 5*), the optimal stoichiometry of proteins in complexes may be perturbed (*Figure 7A*). Indeed, using curated lists of protein complexes (*Cherry et al., 2012*), we found

that an increased deviation from the original stoichiometry occurred with aging (*Figure 7B–D*, and *Figure 7—figure supplement 1–3*). We observed many complexes that were not previously implicated in aging to be age-affected, and we found previously implicated protein complexes such as the vacuolar adenosine triphosphatase (*Hughes and Gottschling, 2012*) and the nuclear pore complex (*Lord et al., 2015*; *Denoth-Lippuner et al., 2014*) to lose stoichiometry (*Figure 7C and D* and *Figure 7—figure supplement 1,2*). The global stoichiometry loss was greater in aged mothers compared with the daughter population (*Figure 7—figure supplement 3A*), confirming that this is an aging-related phenotype. Additionally, we found that the stoichiometry loss was greater overall at the proteome level than at the transcriptome level (*Figure 7B*), supporting the observation that protein levels uncouple from their transcript levels.

Being built of fewer genes (1494 proteins versus 4904 transcripts), the high-level directional network of the proteome was less revealing than that of the transcriptome (*Figure 7—figure supplement 4*). The most causal cluster of the proteome network was enriched for chaperone proteins, reflecting a cellular response to internally changing conditions. Such conditions could include metabolic restructuring in response to an increased cell size or to aggregating proteins that are

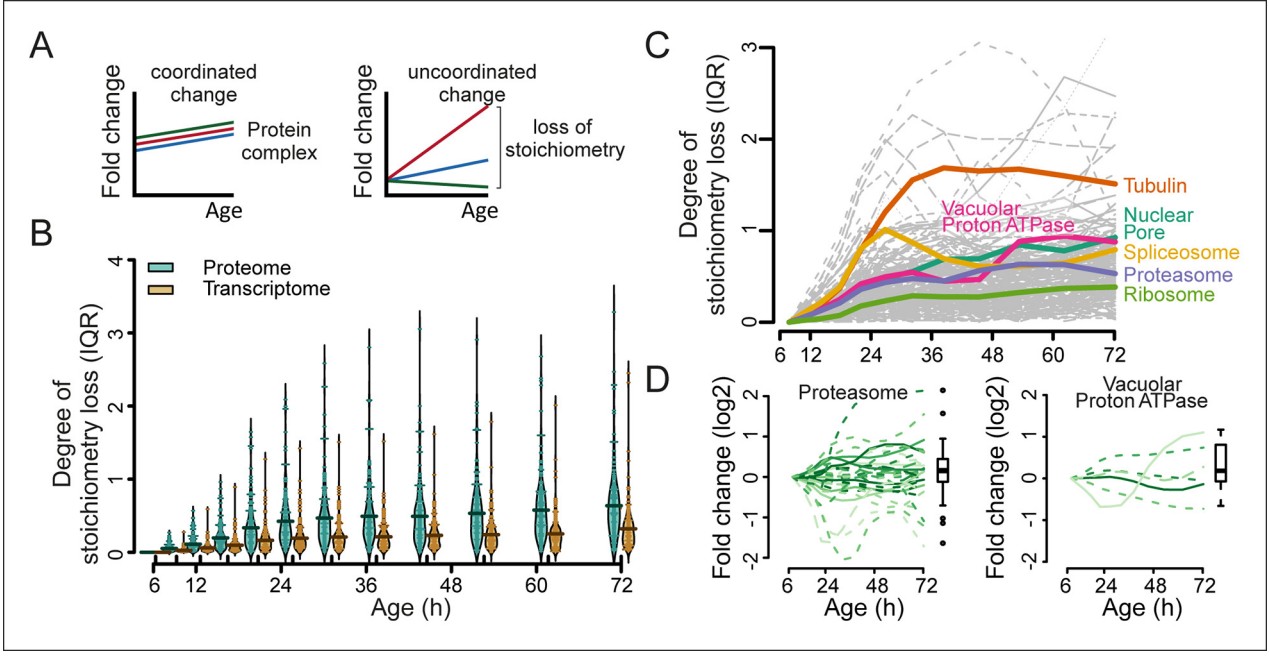

**Figure 7.** Loss of stoichiometry in protein complexes is a consequence during aging. (A) Illustrative representation of loss of stoichiometry within a protein complex during aging. Changing levels of proteins may be coordinated (left) or uncoordinated and result in a loss of complex stoichiometry (right). (B) Stoichiometry loss (for a single complex defined as the InterQuartile Range (IQR) of the distribution of fold changes of the components) is plotted for all complexes in proteome and transcriptome datasets as bean plots during aging. Thick horizontal line represents the mean of the distribution of all complexes, thin colored lines the individual complexes' stoichiometry loss, and the outline the distribution of all complexes. The genes in common between the proteome and transcriptome datasets are used. (C) Illustration of the loss of stoichiometry of protein complexes during aging for the proteome (gray lines), with specific examples highlighted (colored lines). (D) Illustration of the loss of protein stoichiometry in proteasome (left panel) and the vacuolar proton transporting V-type adenosine triphosphatase (ATPase), V1 domain (right panel). The protein abundance changes (log₂ scale) of the complex' components are plotted in time. The degree of stoichiometry loss is indicated with a box plot.

The following figure supplements are available for Figure 7:

**Figure supplement 1.** Proteome data of distribution of changes within complexes in the cell.

**Figure supplement 2.** Transcriptome data of distribution of changes within complexes in the cell.

**Figure supplement 3.** Loss of stoichiometry occurring in the protein complexes.

**Figure supplement 4.** The proteome network.

accumulating due to altered protein complex stoichiometry. Furthermore, we found that the causal clusters of the proteome network tended to be expressed according to their transcriptional message (i.e. coupled expression; Q1, Q3), while the responsive clusters represented increasingly uncoupled expression (Q2, Q4) (*Figure 6—figure supplement 2C*). This both confirmed the response of the cell to the accumulating changes occurring during aging and indicated that the effects of uncoupled protein expression are progressive over time. We see the clear downstream consequences during aging emerging in the proteome, including metabolic shifts, stoichiometric loss, aggregating proteins, and protein overproduction. All of these point to pathways and processes that may become dysfunctional with aging, any of which may ultimately result in cell death.

## Discussion

Using our newly developed culturing and computational methods and state-of-the-art proteomics and transcriptomics analyses, we generated the first systems-level molecular phenotype of replicatively aging yeast. The comprehensiveness of the data allowed us to discover that protein biogenesis machinery genes, including ribosome, tRNA synthesis, and translation regulation genes, have their protein levels uncoupled from their mRNA levels during aging (*Figure 5B*). Furthermore, the dynamic nature of the data allowed us to pinpoint the transcripts of these genes as having the strongest ability to predict the behavior of others transcripts during aging (*Figure 6B*). Finally, we observed metabolic changes, protein stress responses, and changes in the stoichiometry of many protein complexes (*Figure 3B,4,7B*).

Based on these analyses, we propose a model whereby the uncoupling of protein levels of biogenesis-related genes from their transcript levels is causal for the changes occurring in aging yeast. The model proposes that proteins of the translation machinery that are uncoupled from transcript levels accumulate in cells with age (*Figure 5B*). As the biogenesis genes are themselves involved in translation, their uncoupling might contribute to further uncoupling of the proteome from the transcriptome as a whole. This general uncoupling has degenerative effects (i.e. cell size increase, protein aggregations and loss of stoichiometry in protein complexes), that stimulate transcriptional responses in the cell (i.e. metabolic changes and activated stress responses), which further contributes to changes in the proteome. Although we cannot exclude the possibility of other causes even further upstream, the uncoupling of the protein biogenesis machinery is likely an early driver of replicative aging in yeast.

A question remains as to why the biogenesis-related class of proteins we identified as having protein levels uncoupled from their transcript levels become over-represented in replicatively aging yeast in the first place. Ribosome footprinting has shown these proteins to be highly translated (*Ingolia et al., 2009*), and protein turnover experiments have shown them to be highly stable (*Belle et al., 2006*); thus, it is possible that their overabundance may result from the combination of the dynamics of protein biogenesis, protein turnover, and mRNA stability. Interestingly, the ribosomal proteins themselves showed a low degree of loss of stoichiometry at the protein complex level in our data (*Figure 7C*), supporting the idea that they are still active and contributing to uncoupling in the cell. In any case, the uncoupling of protein and transcript levels has downstream consequences for the cell that may explain many phenotypes of aging. First, cell size may increase due to protein overproduction and result in metabolic changes. Second, proteins being overproduced at different rates will alter protein complex stoichiometry. Many documented phenotypes of aging may result from this, including the formation of protein aggregates (*Erjavec et al., 2007*), increased reactive oxygen species formation by a dysfunctional mitochondrial transport chain (*Laun et al., 2001*), and loss of gene silencing (*Hu et al., 2014*). The sum of these may ultimately lead to system failure for the organism.

Directly targeting certain failing protein complexes or downstream deleterious effects results in replicative lifespan extension, but we suggest that many of these effects will prove to be cell type- and growth condition-specific. Our model predicts that a more robust extension of lifespan may be possible in many organisms by targeting the causal factor in aging, protein biogenesis. Indeed, altering the rates of protein production (i.e. translation) or degradation (i.e. autophagy) have repeatedly been shown to influence longevity across a wide range of organisms (see (*Wasko and Kaeberlein, 2014*; *Johnson et al., 2013*; *Cuervo, 2008*). The translation activators, target of rapamycin (TOR) and S6 kinase, fall into this category, and decreases in their activity result in

increased lifespan in yeast (*Fabrizio et al., 2001*; *Kaeberlein et al., 2005*), worms (*Vellai et al., 2003*; *Pan et al., 2007*), flies (*Kapahi et al., 2004*), and mice (*Lamming et al., 2012*; *Selman et al., 2009*), as does calorie restriction and drugs such as rapamycin, which are also modulators of protein biogenesis pathways (*Johnson et al., 2013*). Likewise, deletions in ribosomal subunit components have positive effects on lifespan in both yeast (*Steffen et al., 2008*) and worms (*Hansen et al., 2007*). Our model suggests why these interventions and mutations have a lifespan-extending effect in a broad spectrum of organisms, namely because protein biogenesis machinery is itself a driver of aging.

## Materials and methods

### Aging yeast

Strains and medium

The prototrophic *Saccharomyces cerevisiae* strain YSBN6 (MATa) was used for the phenotyping of yeast replicative aging (*Canelas et al., 2010*). The cells were grown in yeast nitrogen base (YNB) without amino acids (ForMedium, Norfolk, UK) supplemented with 2% glucose at a temperature of 30°C, unless indicated differently. Precultures in flasks were shaken at 300 revolutions per minute (RPM).

Samples not processed by the steps involving biotinylation and the attachment of beads (termed 'unprocessed samples') were precultured in the above medium for minimum 24 hr in mid-exponential growth phase and were immediately pelleted (5 min, 2500 × g) and snap frozen in liquid nitrogen.

Preparing the cells for column captured culturing in aging columns

Prior to loading the cells onto the aging columns, the cells were biotinylated and labeled with iron beads (*Figure 1—figure supplement 1*) in a manner adapted from (*Smeal et al., 1996*), as follows: The yeast YSBN6 was pre-cultured for minimum 24 hr in a mid-exponential growth phase, having an optical density below 1. Cells were harvested and concentrated by gentle centrifugation, 10 min 2500 ×g. For one column, $3 \times 10^9$ cells were resuspended in 1 ml 2× phosphate buffered saline (PBS), immediately mixed with 14 mg Sulfo-NHS-LC-Biotin (Thermo Scientific, Rockford, IL) dissolved in 1 ml cold (4°C) water and incubated in a shaker (800 RPM) at room temperature for 20 min. The biotinylated cells were washed twice with 1×PBS at room temperature and were resuspended in 100 ml pre-warmed YNB plus 2% glucose and incubated for 90 min at 30°C shaken at 300 RPM. At room temperature, the cells were pelleted by gentle centrifugation (5 min, 2500 ×g), washed with 1× PBS, resuspended in 4 ml 1× PBS, mixed with 750 μl of streptavidin coated BioMag beads (Qiagen, Germantown, MD) and incubated for 30 min on a lab rocker. The bead-labeled cells were concentrated in ~~0.5 ml PBS by gentle centrifugation (5min, 2500 ×g) and $2 \times 10^9$ cells were loaded onto the magnetized aging column.

The aging columns setup

The aging column setup is a closed system, where cells are cultivated on a magnetized iron meshwork under a constant flow of medium (*Figure 1—figure supplement 1B and C*). The setup was designed to ensure a sterile environment within the system, continuous removal of daughter cells, and constant oxygen and nutrient concentrations in the medium. Table S1 (*Figure 1—source data 1* shows materials used for its construction and operation.

The core of the setup for column-captured cell cultivation is the 0.3" Negative Selection Column combined with a 3-way stopcock (Stemcell Technologies Inc., Vancouver, Canada; *Figure 1—source data 1*, Table S1), which is placed in a magnetic field. Four magnets (StemSep Red Magnet, Stemcell Technologies) were placed in a stand (custom made, *Figure 1—figure supplement 1*), and four stands with magnets were connected in a row to run 16 columns simultaneously. The rim at the top of the column was cut with a sharp scalpel, to enable connection with 15 cm long silicone tubing (Si, inner diameter (id) 8 mm, outer diameter (od) 11 mm, Si 8-11, *Figure 1—figure supplement 1C*, *Figure 1—source data 1* Table S1). Silicone tubing was chosen, as it is air permeable. The T-

connector (od 10 mm, C T-10) on top serves to connect the column with the inlet tubing from the side and a 6 cm long tubing closed with a clamp (C. II).

The pump (BVP standard motor, MS/CA4-12 + 3 × MS/CA4-12 extensions; Ismatec, Wertheim, Germany) provided a constant medium flow over the column. The pump tubing (BPT Tubing, 1.52 mm ID, 400 mm length, Pharmed, Ismatec, Wertheim, Germany) connected the 20 l medium jar (20 l round high density polyethylene [HDPE] bottle, Nalgene, Rochester, NY) to the column via two long pieces of 2 m silicone tubing (id 2 mm, od 4 mm, Si 2-4). The Silicone tubing between pump and column could be closed with clamp C. II. The flow rate of medium over the column was set at 170 ml/h.

The medium jar was closed with a 5-layered aluminum foil top prior to autoclaving. 5 syringes with their plungers removed were punched through the aluminum foil and 4 were connected inside the jar to a 60 cm long silicone tubing (id 6 mm, od 8 mm, Si 6-8). The end of the tubing was weighted down with a glass pipet, in order to have the inlet remain at the bottom of the jar. The syringe barrels at the top of the jar were closed with small pieces of aluminum foil during autoclaving and attached to the Si 2-4 silicon inlet tubing prior to the start of the column run. The fifth syringe without its plunger and without silicon tubing was attached on the outside to Si 2-4 silicone tubing, with pressurized sterile air, to provide an overpressure of sterile air in the medium jar. The medium jar was filled with 20 l autoclaved YNB without amino acids prior to autoclaving and was subsequently supplemented with 2% filter-sterilized glucose.

The effluent of the column went down via silicone Si 2-4 tubing, passing a quick release connector, and went up via silicone Si 4-6 tubing to an air chamber. The tubing could be closed with a clamp (C. III). The air chamber breaked the laminar medium flow, allowing the liquid to drip down via silicone Si 4-6 tubing into a waste jar (20 l round HDPE bottle, Nalgene). The air chamber consisted of a T-connector (od 10 mm, C T-10) connected at all the three sides with 6 cm silicone Si 8-11 tubing and a tube connector.

## Loading the aging columns

Prior to loading the columns with the biotinylated yeast cells, the system was primed with sterile medium for about 1 hr, having clamp C. I and C. III open. The medium flow was then stopped on the pump and clamps C. I and C. III were closed and clamp C. II opened. The quick release was opened and clamp C. III was shortly opened to lower the medium level to the iron meshwork. The column was detached from the tubing and the magnet and $2 \times 10^9$ cells were pipetted onto the column and gently sucked into the meshwork by a 5 ml syringe attached to the stopcock below the column. The stopcock was closed, ~2ml fresh medium was pipetted on top of the column and the column was reattached to the tubing and placed in the magnet. Clamp C. I was opened and the medium flow was restarted. After some medium was collected on top of the column, clamp C. III was opened. Clamp C. II was kept open until the medium level above the column stabilized halfway in the tubing above the column. This level could be adjusted by the height of the air chamber in the effluent tubing (*Figure 1—figure supplement 1B*). The cells were kept surrounded by liquid media throughout all cultivation time.

## Harvesting aged yeast cells

In order to harvest the mother cells, the pump was stopped, clamps C. I and C. III were closed and clamp C. II opened. Only the specific pump tubing was disconnected from the pump, and the pump was restarted. The quick release was disconnected and through the shortly opening clamp C. III, the medium level was lowered to just above the meshwork. The tubing on top of the column was detached and a 20 ml syringe was connected to the stopcock below the column. While keeping the column at the magnet, 15 ml fresh medium was provided on top of the column, while the column effluent was collected by the syringe. This step was repeated 2 or 3 times, until the effluent was clear. This combined column effluent sample was kept on ice (effluent fraction, sample: Mix 3, *Figure 2—figure supplement 1*). The column was detached from the magnet and again 15 ml fresh medium was provided on top of the column and the effluent was collected by a new syringe. This was repeated 2 or 3 times, until the medium was clear. This combined column fraction (column fraction, later to be split into Mix 1 and 2, *Figure 2—figure supplement 3A*) was also kept on ice.

After harvesting, the samples consisted of mixes of aged mother cells, dead cells, and daughter cells. In order to obtain a higher purity of aged mother cells, an enrichment step was required for

the column fraction. The cells were gently centrifuged (10 min 2500 ×g), resuspended in 7 ml cold PBS and transferred to a glass test tube. The test tube was placed in a magnet ("The Big Easy" EasySep Magnet, Stemcell Technologies Inc. Grenoble, France) for 5 min (*Figure 2—figure supplement 3A*, panel II). The supernatant was removed by pipetting and the magnet-bound cells were resuspended in fresh and cold PBS. This was repeated two times, until the supernatant was clear. The supernatant fractions were combined and kept on ice (wash fraction, sample: Mix 1). The cells that were retained in the magnet were resuspended in 2 ml PBS after removal from the magnet (mother enriched fraction, sample: Mix 2) (*Figure 2—figure supplement 3A*, panel III). The samples were pelleted by gentle centrifugation (4 min, 4°C, 2500 ×g) and immediately snap frozen in liquid nitrogen. A small aliquot of each of three samples was kept aside to measure the fractions of live and dead cells, mother and daughter cells, and obtain the cell count per sample.

## Harvesting time points

Based on the population viability curves generated from the columns during test campaigns, the average lifespan of yeast being roughly 20–30 divisions, and the doubling time of the YSBN6 strain being roughly 2 hr, it was decided to collect aged samples up to 72 hr of aging, with roughly 42% of viable cells expected in the last sample (*Figure 1B*). There is cell-to-cell variation in the replication rates of yeast and so, with time, the distribution of replicative ages per sample increases. These distributions were modeled based on the variation of the replication rates as quantified from single cell microfluidic data (unpublished data). In a mathematical model, a start culture of 1000 cells having a random replication rate, according to a Poisson distribution an average replication rate of 0.5 $hr^{-1}$, was allowed to replicate (and age) (*Figure 1—figure supplement 4*). Consistent with our empirical observations counting bud scars in the population (*Figure 1—figure supplement 2D*), with increasing elapsed time, the distributions of the number of replications per cell became wider. Linearly spaced harvesting in time would cause increasing information overlap between neighboring time points, thus it was decided to harvest samples exponentially spaced in time (*Figure 1—figure supplement 4*).

Finally, we performed two replicate runs of the column-captured cell culturing campaigns. Campaign 1 generated an unprocessed sample and 14 column samples and campaign 2 generated another unprocessed sample and 8 column samples. In total, two unprocessed samples combined with 16 unique time points were generated (*Figure 2—figure supplement 5*).

## Flow cytometry analysis of sample composition

In each sample, the cells were counted on a BD Accuri C6 flow cytometer (Becton, Dickinson and Company, Franklin Lakes, NJ). To quantify the fractions of mother cells, dead cells, and daughter cells in the samples, the cells were stained with dyes and analyzed by flow cytometry using the BD Accuri C6. From each aliquot, $2 \times 10^6$ cells were pelleted and resuspended in 100 µl PBS, and simultaneously stained for 30 min at room temperature with 5 µl 5 mg/ml fluorescein isothiocyanate conjugated Avidin (AvF, Thermo Scientific) and 2 µl 2 mM propidium iodide (PI, Sigma-Aldrich Co., St. Louis, MO). Biotinylated mother cells (see section Materials and methods, Preparing the cells for column-captured culturing in aging columns) were stained with AvF, dead mother or dead daughter cells were stained with PI, live daughter cells remained unstained (*Figure 1—figure supplement 5*). The fluorescein was excited by a laser of 488 nm wavelength and detected in the range of 533 ± 30 nm, PI was excited by a laser of 488 nm and detected in the range of >670 nm. The beads were excluded from any analysis by gating (*Figure 1—figure supplement 5B*, left panels). The flow cytometer events that were plotted for their PI and AvF intensities in a scatter plot, clear clusters for stained and unstained, both in PI and AvF channel, were apparent. The fractional enrichments were obtained in the BD CS Accuri C6 Software 1.0 (*Figure 1—figure supplement 5*).

## Validations of column captured cultivation

*Oxygen concentration in medium:* The oxygen concentration was measured by using the Optical Oxygen Meter Fibox 3 (PreSens Precision Sensing GmbH, Regensburg, Germany). The flow-cell, an oxygen-sensitive spot glued in a polystyrene tube, was connected to the tubing in front of the aging column to measure the $O_2$ concentration in fresh medium and connected to the effluent tubing to

measure the $O_2$ concentration in the column effluent. Each measurement was done within 10 min to avoid measurements being influenced by the accumulation of yeast cells in the flow-cell, which would alter readings.

*Glucose consumption on the column:* The glucose concentration in the medium and the column effluent was measured with a commercially available enzyme-based assay Enzytec fluid D-Glucose (Thermo Fisher Scientific GmbH, Dreieich, Germany). The column effluent samples were harvested by collecting medium from the column outlet, by opening the quick release below the column (*Figure 1—figure supplement 1B and C*). The column effluent sample was immediately placed on ice, shortly centrifuged (30 s, >16 k ×g) to remove the cells, and the glucose concentration was measured.

*Bud scar counting:* The number of bud scars was counted using microscopy and evaluated from flow cytometry data.

For microscopy, $1 \times 10^7$ cells were resuspended in 0.5 ml PBS supplemented with 25 µl 5 mg/ml Alexa 633 labeled wheat germ agglutinin (WGA, Life Technologies/Thermo Fisher Scientific Co., Carlsbad, CA), 50 µl 5 mg/ml AvF and 20 µl 2 mM PI and incubated for 90 min at room temperature (see 'Flow cytometry analysis of sample composition'). The images were taken on a commercial laser scanning microscope Zeiss LSM710 (Carl Zeiss, MicroImaging, Jena, Germany), using ZEN2010B software. The dyes were excited with different solid state lasers; PI and AvF were excited with a wavelength of 488 nm and emission was recorded between 607–797 and 493–564 nm wavelength, respectively; WGA Alexa 633 was excited, with a wavelength of 633 nm and emission was recorded between 638–797 nm wavelength in a stack of 10 images with a z-scaling of 0.8 µm (*Figure 1D*, inset). Only living mother cells were selected (containing AvF, without PI) and the bud scars were counted independently by two researchers.

For flow cytometry, $2 \times 10^6$ cells were resuspended in 100 µl PBS supplemented with 7 µl 5 mg/ml WGA Alexa 633 and incubated for 30 min at room temperature. The cells were excited in the flow cytometer by a laser with 640 nm wavelength and emission was recorded with a filter selecting for $675 \pm 25$ nm. The mean fluorescence intensity for R2 was normalized to R1 t = 0 h, to be plotted on the same scale (*Figure 1D*).

*Lifespan curve:* For viability of mother (AvF positive) and daughter (AvF negative) cells at each time point in the aging column, viability of the mother and daughter cells was assessed in each mixed-cell sample (derived from proportions of live [PI negative] and dead [PI positive] cells (*Figure 1—figure supplement 5*, *Figure 2—figure supplement 3B*). These scores were weighted based on the number of cells present in each of these samples (derived from raw numbers as presented in *Figure 1—figure supplement 3*). This ensured that the viability of mothers and daughters (*Figure 1C*) reflected the entire population, since mothers and daughters in different mixed-cell samples may have slightly different ratios of live to dead cells. The microfluidic-based lifespan curve was obtained from authors of *Huberts et al., 2014*, based on 2641 cells, plotted as viability versus time.

## Proteome analysis

### $^{15}$N standards

Protein extracts from isotopically labeled $^{15}$N YSBN6 yeast cells were used as an internal standard for the targeted selected reaction monitoring (SRM) proteomics experiments. For the preparation of the $^{15}$N standards, yeast was cultivated in two 2.5 l fermenters on minimal or synthetic Verduyn medium (*Verduyn et al., 1992*), supplemented with 10 g/l glucose and using $^{15}$N-labeled $(NH_4)_2SO_4$ as the sole nitrogen source. Cells were harvested in the different growth phases, namely the log phase (L), the deceleration phase (D) and the stationary phase (S, *Figure 2—figure supplement 1A*). Aliquots from all conditions were mixed (1:1:1) to maximize the coverage of the targeted proteins.

### Cell lysis and protein extraction

Cell pellets were resuspended in 1.85 M sodium hydroxide plus 7.4% v/v β-mercapto-ethanol at a concentration of $1 \times 10^8$ cells per 100 µl and incubated for 10 min on ice. An equal volume of 100% w/v trichloric acid (TCA) was added and was subsequently incubated for 10 min on ice. The precipitated proteins were collected by centrifugation (16 k ×g, 10 min, 4°C). The pellet was washed with 200 µl cold acetone and incubated for 30 min at −20°C. Finally, the protein pellet was collected by centrifugation (16 ×g, 10 min, 4°C), and removal of supernatant.

The precipitated proteins were resuspended in 100 µl 2% w/v sodium deoxycholate plus 100 mM ammonium bicarbonate per $1\times10^8$ cells. For the targeted proteomics, the [15]N-labelled protein extracts were added in a 1:1 ratio, based on the cell counts. Samples were incubated for 5 min at 90°C to solubilize. Magnetic beads present in a subset of the samples were removed at this stage by collecting them on the commercially available magnet tube rack DynaMag-2 (Life Technologies/ Thermo Fisher Scientific Co., Waltham, MA, USA).

## Digestion and cleanup

The solubilized proteins were reduced with 12 mM dithiothreitol (30 min at 55°C) and alkylated with 40 mM iodoacetamide (45 min at 30°C, in the dark). Samples were diluted with 100 mM ammonium bicarbonate to dilute the sodium deoxycholate to 1% w/v prior to overnight digestion with trypsin (1:100, sequencing grade modified trypsin V5111, Promega, Madison, WI) at 37°C. Then, 10% v/v formic acid (FA) was added to the solution to precipitate the deoxycholate, which was subsequently removed by centrifugation (16 k ×g, 10 minutes). Cleanup prior to liquid chromatography–mass spectrometry (LC–MS) analysis was done with C18-SPE columns (SPE C18-Aq 50 mg/1ml, Gracepure, Columbia, MD). This column was conditioned with 3 × 1 ml acetonitrile (ACN) plus 0.1% v/v FA, and re-equilibrated with 3 × 1 ml 0.1% v/v FA before application of the samples at a total amount of maximum 1 mg total protein per column. The bound peptides were washed with 2 × 1ml 0.1% v/v FA and eluted with 3 × 0.4 ml 50% v/v ACN plus 0.1% v/v FA. The eluted fractions were dried under vacuum and resuspended in 0.1% v/v FA to a final concentration of around 1 µg/µl.

## Targeted proteomics (SRM)

SRM analyses were performed on a triple quadrupole mass spectrometer with a nanoelectrospray ion source (TSQ Vantage, Thermo Fisher Scientific, Waltham, MA). Chromatographic separation of the peptides was performed by liquid chromatography on a nano ultra-high performance liquid chromatography system (UltiMate UHPLC Focused, Dionex, Thermo Fisher Scientific, Waltham, MA) using a nano column (Acclaim PepMap100 C18, 75 µm x 150 mm 3 µm, 100 Å, Dionex, Thermo Fisher Scientific, Waltham, MA). Samples were injected at a total amount of 1 µg using the microliter-pickup system using 0.1% v/v FA as transport liquid from a cooled autosampler (5°C) and loaded onto a trap column (µ-precolumn cartridge, Acclaim PepMap100 C18, 5 µm, 100 Å, 300 µm id, 5 mm Dionex, Thermo Fisher Scientific, Waltham, MA). Peptides were separated on the nano-LC column using a linear gradient from 3 to 45% v/v ACN plus 0.1% v/v FA in 30 min at a flow rate of 0.3 µl/min. The mass spectrometer was operated in the positive mode at a spray voltage of 1500 V, a capillary temperature of 270°C, a half maximum peak width of 0.7 for Q1 and Q3, a collision gas pressure of 1.2 mTorr and a cycle time of 1.2 ms. The measurements were scheduled in windows of 4 min around the pre-determined retention time, with a maximum of 150 concurrent transitions.

The MS traces were manually curated using the Skyline software (*Steffen et al., 2008*). The sum of all transition peak areas for the endogenous and standard ([15]N labeled) peptide was used to calculate the ratio between the endogenous and standard peptides. Only peptides that were minimally quantified with two transitions and a peak area of the [15]N standard above 10,000 for both technical replicates were considered for quantification. The ratios on protein level were calculated by averaging the ratio of all peptides per protein. In order to correct for global errors made in the protein concentration determination of either the endogenous samples or the [15]N labeled standard, the median of all datasets were normalized to the same value.

## Shotgun proteomics

1 µg of peptides of each sample were subjected to LC–MS analysis using a dual pressure LTQ-Orbitrap Velos mass spectrometer connected to an electrospray ion source (Thermo Fisher Scientific, Waltham, MA) as described recently (*Sun et al., 1994*) with a few modifications. In brief, peptide separation was carried out using an EASY nLC-1000 system (Thermo Fisher Scientific, Waltham, MA) equipped with a reversed phase HPLC column (75 µm × 45 cm) packed in-house with C18 resin (ReproSil-Pur C18–AQ, 1.9 µm resin; Dr. Maisch GmbH, Ammerbuch-Entringen, Germany) using a linear gradient from 95% solvent A (0.15% FA, 2% acetonitrile) and 5% solvent B (98% acetonitrile, 0.15% FA) to 28% solvent B over 120 min at a flow rate of 0.2 µl/min. The data acquisition mode was set to obtain one high resolution MS scan in the Fourier Transform (FT) part of

the mass spectrometer at a resolution of 60,000 full width at half-maximum (at m/z 400) followed by MS/MS scans in the linear ion trap of the 20 most intense ions. The charged state screening modus was enabled to exclude unassigned and singly charged ions and the dynamic exclusion duration was set to 30 s. The ion accumulation time was set to 300 ms (MS) and 50 ms (MS/MS).

For label-free quantification, the generated raw files were imported into the Progenesis LC-MS software (Nonlinear Dynamics, Version 4.0) and analyzed using the default parameter settings. MS/MS-data were exported directly from Progenesis LC–MS in mgf format and searched against a decoy database the forward and reverse sequences of the predicted proteome from *S. cerevisae* (SGD, download date: 15/6/2012, total of 13,590 entries) using MASCOT (version 2.4.0). The search criteria were set as follows: full tryptic specificity was required (cleavage after lysine or arginine residues); three missed cleavages were allowed; carbamidomethylation (C) was set as fixed modification; oxidation (M) as variable modification. The mass tolerance was set to 10 ppm for precursor ions and 0.6 Da for fragment ions. Results from the database search were imported into Progenesis and the final peptide feature list and the protein list containing the summed peak areas of all identified peptides for each protein, respectively, were exported from Progenesis LC-MS. Both lists were further statically analyzed using an in-house developed R script (SafeQuant) and the peptide and protein false discovery rate (FDR) was set to 1% using the number of reverse hits in the dataset (*Sun et al., 1994*).

## Transcriptomics

### mRNA extraction

For the extraction of mRNA from yeast, the RiboPure RNA Purification Kit, yeast (Ambion, Life Technologies/Thermo Fisher Scientific Co.) was used as described by the manufacturer. Frozen cell pellets of $3 \times 10^7$ cells were suspended in the lysis mixture. Vortexing was done by using the the Ambion Vortex Adapter (Ambion, Life Technologies/Thermo Fisher Scientific Co. Waltham, MA). The mRNA was collected in 70 μl elution solution. The quality and yield of the RNA was checked with a NanoDrop ND-1000 Spectrophotometer (Thermo Fisher Scientific, Waltham, MA). The samples were stored as 5 μg mRNA aliquots at −80°C. 1 μl of 1:10 diluted mixture of 92 polyadenylated non-yeast transcripts was added as a spike-in for sequencing quality control (ERCC RNA Spike-In control mix, Life Technologies/Thermo Fisher, Waltham, MA) (*Smeal et al., 1996*).

### mRNA sequencing and mapping

The mRNA was sequenced by ServiceXS (Leiden, The Netherlands). The quality and integrity of the RNA samples was determined with a Nanodrop ND1000 spectrophotometer and analyzed on a RNA 6000 Lab-on-a-Chip using bioanalyzer (Agilent Technologies, Santa Clara, CA). The complementary DNA (cDNA) libraries were generated by using the Illumina TruSeq mRNA-Seq Sample Prep Kit v2 (Illumina, San Diego, CA). In short, mRNA was isolated from total RNA using the oligo-dT-magnetic beads and fragmented and cDNA synthesis was performed. The cDNA was ligated with the sequencing adapters and amplified by polymerase chain reaction. The quality of the amplified cDNA was measured with a DNA 1000 Lab-on-a-Chip. The fragment sizes ranged between 300 and 500 bp.

The cDNA was clustered in the flow cell of the sequencer by an Illumina cBot and the sequencing was done on an Illumina HiSeq 2000. A cDNA concentration of 4.5 pM was used for sequencing, in two reads of 100 cycles each, controlled by the HiSeq control software HCS v2.0.12.0. Image analysis, base calling, and quality checks were performed with the Illumina data analysis pipeline RTA v1.13.48 and/or OLB v1.9 and CASAVA v1.8.2. All data consisted of >0.9 Gb read depth and a quality Q30-score >80% per sample. One time point set, replicate 1 t10 (26.8 hr), was excluded by this criteria.

Reads were mapped to EF4 genome assembly using TopHat software v2.0.8 and gene annotation from Ensembl release 71. Per gene expression values were calculated using Cufflinks/Cuffdiff package v. 2.1.1. Data quality was assessed by principle component analysis on the resulting raw data of spike-in controls and gene profiles. Outliers resulting from poor sequencing results in the spike-in (i.e., *Figure 2—source data 2* Table S3.1, samples from replicate 1: t2_M_Feb and t7_EW_Feb) or the full genome profiles (i.e., *Figure 2—source data 2* Table S3.1, samples from replicate 2: t14_M_May, t14_D_May and t14_EW_May) were removed. As a result, three time points were

omitted: replicate 1 t2 (1 hr), replicate 1 t7 (14 hr), replicate 2 t14 (53 hr). In total, four time points were omitted from the raw mRNA data.

## Data processing

### Mathematical un-mixing

Mathematical un-mixing rests on the idea that a system of linear equations can be solved when (i) the number of equations is equal to the number of unknowns and (ii) these are independent (see supplemental note 1 in *Supplementary file 1* for terminology, explanation, and validation of the method). In our experiment, this idea was implemented for each time point by means of a weighted 'un-mixing' matrix (**W**) whose rows represented the fractions of cell types (i.e. mothers, dead, and daughter cells) in the harvested 'mixed-cell samples'. The fractional composition of each mixed-cell sample was acquired by using flow cytometry on dye stained cells, using PI and AvF to assess the amount of live mother cells, dead cells, and daughter cells (see: Materials and methods, Flow cytometry analysis of sample composition, and see *Figure 2—source data 1* Table S2.f for each time point's matrix).

For the mathematical un-mixing validation experiments, the fractional compositions of the mixed-cell samples were defined by mixing different pure cell sample types (i.e. log-phase, deceleration-phase, and stationary-phase cells) in known ratios. Protein and mRNA abundance values for the mixed-cell samples were measured by targeted (SRM) proteomics (for validation only) and shotgun proteomics (for validation and aging cells), or RNAseq transcriptomics (for validation and aging cells). Equation (3), present in the supplemental note 1 in *Supplementary file 1*, was implemented using a custom R script for the actual un-mixing procedure. Following the un-mixing of the data, should the resulting data contained 'unsolvable' entries (see supplemental note 1 in *Supplementary file 1*), a data quality criteria was applied: at least five time points per time trace (0–72 hr) should be solvable, otherwise the protein or transcript was removed from the dataset. In cases that passed this criteria but still contained one or more unsolvable entries in the time series, the missing data was linearly interpolated by the time points neighboring the data in question using the 'approx' interpolation function in R, implemented by the zoo package (*Zeileis and Grothendieck, 2005*). Datasets were subsequently normalized to 1 million for both the shotgun proteomes and transcriptomes.

### Correction for effect of beads

A simple correction step accommodating for the specific protein losses caused by the presence of the beads was applied to the relevant data, and is explained in the supplemental note 2 in *Supplementary file 1*. The loss was specific for each protein, highly reproducible and independent of the ratio of beads to cells (*Figure 2—figure supplement 4*). Briefly, a protein specific correction factor was calculated for each protein of the proteome from the difference between a sample with and without beads, averaged over two replicates. The correction was applied to the raw proteome datasets, prior to mathematical un-mixing, and on all samples that contained beads.

### Selection of the young time point reference sample

A young time point to compare aged cells to was selected and processed as described in the supplemental note 3 in *Supplementary file 1*. Briefly, the time series proteome and transcriptome data were standardized to the difference between the starting time point (7.8 hr in the column) and an unprocessed sample, and only data from 7.8 hr and later was considered in the analyses. This was done to avoid mislabeling any biological recovery from the biotinylation and loading procedure as being aging related and to maintain quantitative datasets for analysis.

### Data fitting and filtering

For both the shotgun proteomes and transcriptomes, replicate datasets were fitted with a locally weighted scatterplot smoothing by polynomial regression using a standard span value of 0.75 (R Core Development Team, 2014), using the replicates of unprocessed samples, and the replicate time series of 7.8–72 hr, as input for the regression. Final datasets were generated by resampling the regression fit at each time point physically sampled in the experiment (including those prior to 7.8 hr, for completeness and consistency). Datasets are available in Table S2.5a and S3.5a for each

of the proteome and transcriptome supplementary Tables (*Figure 2—source data 1 and 2*). A noise threshold was applied to the time series datasets using the coefficient of variation between replicates with a cutoff of 0.3, corresponding to retention of 90.9 and 84.4% of the most reproducible data for the proteome and transcriptome, respectively (*Figure 2—figure supplement 6*) (dataset available in *Figure 2—source data 3 and 4*, Table S4 and S5). From this final dataset of 1494 proteins and 4904 transcripts, two proteins and two transcripts contained a negative data point in their time series profiles, and were removed from both mother and daughter datasets in subsequent analyses. Unless specified otherwise (see network methods), the final datasets used for analyses consisted of the fitted regression data (*Figure 2—source data 3 and 4*, Table S4 and S5), from 7.8 hr of cultivation and later.

## GO term selection and annotation

Gene functional enrichments were determined by using the DAVID Bioinformatics Resources version 6.7 (*Huang et al., 2009*). Corresponding background gene lists of indicated size (*Figure 3—source data 1* Table S6) were used for each enrichment analysis. Annotation clusters determined by DAVID (groupings of related genes based on the agreement of sharing similar annotation terms) having an enrichment score of >0.5 were selected for consideration, if a GO term was enriched in the cluster with a *p*-value <0.1. For larger datasets, a more stringent enrichment score cutoff of either >0.9 or >1.0 was used, as seen from lowest score cutoffs listed in the table below per analysis. A representative naming for the enrichment was selected after evaluation of the annotation cluster's GO terms (see *Figure 3–source data 1* ,Table S6). Visualization of the representative terms in clouds was made using the R wordcloud package (*Fellows, 2014*) using the annotation cluster enrichment score as a size-scaling factor. If duplicate terms were present within a GO term enrichment list, the higher enrichment was used for visualization purposes. In one instance (the most responsive cluster of the proteome network) an unclear term ('BNR repeat') representing three genes was omitted even though it passed our criteria for inclusion.

## Protein complex deregulation

A curated list of protein complexes derived from the 'cellular component' GO was downloaded from yeastgenome.org (*Cherry et al., 2012*). Using the fold changes of gene products (i.e. either proteins or transcripts) at any given time point within a protein complex of interest, the degree of deregulation was assessed by measuring the interquartile of the distribution of the fold changes of the complex's gene products.

## Network analysis

To infer the high-level directional networks (*Figure 6B*, *Figure 7—figure supplement 1*) and find causal relations, six data analysis steps (*Figure 6—figure supplement 1*) were undertaken, as expanded upon below in the supplemental note 4 in *Supplementary file 1*. Briefly, these were: (i) Starting from the replicate datasets, the gene expression time series of both the transcriptome and proteome were filtered to remove flat and/or noisy profiles using the R package GPREGE (*Zeileis and Grothendieck, 2005*). (ii) The gene product networks (i.e transcriptome or proteome) were generated, based on the gene profiles of the respective time course datasets, using the R package GeneNet (*Opgen-Rhein and Strimmer, 2007*; *Schaefer et al., 2015*). This included generating an undirected network by calculating the partial correlation among gene profiles (*Figure 6—figure supplement 1B and C*, *Figure 7—figure supplement 4A*). (iii) Following this, a directed network was generated from the undirected network, based on an assessment of a gene profile's ability to predict another gene profile (*Figure 6—figure supplement 1D and E*, *Figure 7—figure supplement 4A*) (*Opgen-Rhein and Strimmer, 2007*; *Schaefer et al., 2015*). (iv) The nodes in the network were clustered together, using the method in (*de Magalhães et al., 2012*) using the R package igraph (*Figure 6A*, *Figure 7—figure supplement 4A*) (*Csárdi et al., 2015*). The causal in/out connections among genes were calculated for all the network clusters and listed in a direction matrix (listed in *Figure 6—source data 1* Table S7). (v) A high-level directional network was generated, where the clusters were plotted in the order of their causal ranking by drawing the direction matrix as arrows between the clusters. (vi). A sensitivity analysis was made to determine the optimal sparsity of the networks and the cut-off for the partial correlation among gene profiles.

## Acknowledgements

We thank Hjalmar Permentier, and members of the Veenhoff, Chang, and Heinemann laboratories for constructive discussions. We thank Yves Barral, Peter Lansdorp, Michael Chang, and Christian Riedel for critical review of the manuscript. We thank Daphne Huberts for providing the time-based microfluidic lifespan curve of strain YSBN6. We also thank members of the Heinemann and Poolman laboratories and Stem Cell Technologies for their practical help with the experiments. This project was funded by the Netherlands Organization for Scientific Research (NWO; Systems Biology Centre for Energy Metabolism and Aging).

## Additional information

### Funding

| Funder | Grant reference number | Author |
| --- | --- | --- |
| Nederlandse Organisatie voor Wetenschappelijk Onderzoek | 853.00.110 | Rainer Bischoff<br>Ernst C Wit<br>Liesbeth M Veenhoff<br>Matthias Heinemann |

The funders had no role in study design, data collection and interpretation, or the decision to submit the work for publication.

### Author contributions

GEJ, ACM, Conception and design, Acquisition of data, Analysis and interpretation of data, Drafting or revising the article; JG, Analysis and interpretation of data, Drafting or revising the article; JCW, Acquisition of data, Analysis and interpretation of data; AS, RB, Acquisition of data; VG, ECW, Analysis and interpretation of data; LMV, MH, Conception and design, Analysis and interpretation of data, Drafting or revising the article

## Additional files

### Supplementary files

• Supplementary file 1. Supplementary text and supplementary references (*1–13*).

### Major datasets

The following datasets were generated:

| Author(s) | Year | Dataset title | Dataset ID and/or URL | Database, license, and accessibility information |
| --- | --- | --- | --- | --- |
| Janssens GE, Meinema AC, González J, Wolters JC, Schmidt A, Guryev V, Bischoff R, Wit EC, Veenhoff LM, Heinemann M | 2015 | E-MTAB-3605 - Yeast transcriptome profiling in replicative ageing | https://www.ebi.ac.uk/arrayexpress/experiments/E-MTAB-3605/ | Publicly available at the EMBL-EBI ArrayExpress (Accession no: E-MTAB-3605). |
| Janssens GE, Meinema AC, González J, Wolters JC, Schmidt A, Guryev V, Bischoff R, Wit EC, Veenhoff LM, Heinemann M | 2015 | Aging Yeast | http://www.ebi.ac.uk/pride/archive/projects/PXD001714 | Publicly available at the EMBL-EBI PRIDE Archive (Accession no: PXD001714). |

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
