## [Decision Letter]

Thank you for submitting your work entitled "Protein Biogenesis Machinery is a Driver of Replicative Aging in Yeast" for peer review at *eLife*. Your submission has been favorably evaluated by Naama Barkai (Senior Editor), Karsten Weis (Reviewing Editor), and two reviewers.

The reviewers have discussed the reviews with one another and the Reviewing Editor has drafted this decision to help you prepare a revised submission.

This manuscript describes a novel column-based mother enrichment method to map dynamic transcriptome and proteome changes during the yeast replicative aging process. The authors perform a careful computational analysis to separate the contribution from mother, daughter and dead cells at each time point, and they discover new molecular changes that have not been previously identified. In particular, they find a progressive decoupling of the proteome from the transcriptome for biogenesis genes during aging.

There was a consensus amongst the reviewers that this work describes interesting and important observations and that this manuscript could be published in *eLife* upon appropriate revision. In particular, the reviewers felt that some of the key conclusions are overstated and need to be toned down, and that further experiments are needed to strengthen the results.

Essential revisions:

*Reviewer 1:*1) The authors contend that they have isolated late age mothers. In Figure 1 they show a progressive decline in viability. How do they know that cells didn't die simply because they stayed longer in the column? They rely on additionanl criteria, but that data appears less convincing. Staining for bud scars provides the replicative age, and they track that with WGA intensity (Figure 1). However, the difference is ~ 2-fold, from start to finish. Why doesn't the signal increase more? One would expect that by 72h most cells would have >20 scars, while close to the start only ~2-3 scars. Likewise, cell size increases {less than or equal to} 2-fold (Figure 1). Others reported that the size increase is ~ 4-fold (DOI: 10.4161/cc.10.1.14455). Addressing these issues is important, for their thesis that properly aged cells have been isolated.

2) They report a strong un-coordination between the transcriptome and the proteome, getting worse in later stages. They obtained data for 1494 proteins and 4904 mRNAs. Hence, they did not sample >1/2 of the proteome. Still, sampling 1494 gene products represents a substantial dataset. My concerns have to do with their transcriptome/proteome comparisons. It has been argued that log2 comparisons (which is what they are doing) maybe prone to artifacts (PNAS, 107: 21487). A comprehensive comparison of available proteome, transcriptome and ribosome profiling data in yeast found widespread errors due to noisy data and incomplete coverage (DOI: 10.1371/journal.pgen.1005206), accounting for most of the discrepancies between mRNA and protein abundance. The data here are internally controlled, since the same methods are applied throughout the time series. Nonetheless, since proteome/transcriptome discrepancies are central, it will be prudent to analyze their data with different statistical models, and report on the overlap. They should also report how their findings for the early time points (which should be very similar to populations sampled previously) stack up against the literature.

3) They conclude that changes in the translation machinery are causal to the aging process. This may be true, and fits well with the literature. However, as I understand it, they exclusively rely on the directionality of the network they have built (shown in Figure 5). For that, they look at whether the profile of a given gene product could predict the profile of another gene product. The ones that could *not* be predicted were clustered as causal. Hence, the approach appears to rely on *negative* evidence (lack of predictability) to arrive at such a powerful conclusion. For such a conclusion, I think some additional kind of evidence must be provided. At a minimum, they should look at the numerous mutations (mostly loss-of-function, but some gain-of-function) that have been reported to extend longevity in yeast. What is the enrichment for these gene products in their various clusters? Also, can they point to some new, unidentified ones from their study and test directly their role in determining replicative lifespan?

*Reviewer 2:* 1) The authors found that the biogenesis genes are enriched in the gene cluster, the transcriptional changes of which are "most causal" (ranked number 1) to other transcriptional changes during aging. They then immediately claim in the subsection “Network interference identifies protein biogenesis related genes as casual in yeast aging” that "Taken together, as the biogenesis transcripts were the most predictive of all the changes we measured, our analyses support a model whereby the uncoupling of protein levels of biogenesis-related genes from their transcript levels is causal for the changes occurring in aging yeast". We think the authors need to better explain their reasoning here, because it is a key claim in the paper.

Specifically, we would like to understand the following: (a) Is the decoupling of protein levels from transcript levels of biogenesis genes the causal factor for the transcriptiome changes, or the changes of both transcriptome and proteome, or their global decoupling during aging? (b) What is the mechanism for the decoupling to cause transcriptome changes, since it appears that the authors came up with this claim from their transcriptional network analysis? (c) If the authors want to claim that the decoupling of protein levels from transcript levels of biogenesis genes can cause the global decoupling of the proteome and transcriptome during aging (with which we are totally fine), then the best evidence would seem to arise from a similar network analysis for the decoupled time series of the decoupled genes (Q2 and Q4 in Figure 4). The network correlation analysis of the transcriptome in Figure 5 seems less than ideal in supporting this claim.

To summarize this point, the authors present very rich and impressive data, but we believe that they need to be more specific about what they mean in order to draw such strong conclusions. A titrated summary would be sufficiently interesting: The decoupling of biogenesis protein levels from their transcript levels through some early-life molecular regulation (e.g. a general stress response?) can drive the global decoupling of the proteome and transcriptome, which is a consequence of aging and an important molecular phenotype of aged cells. Here, the overabundance of biogenesis genes is an early event driving a specific molecular consequence of aging, rather than the cause of all the transcriptome and proteome changes during aging.

2) It is stated that only 30% of the cells are alive at the final (12th) time point and the rest have died due to aging. Is it possible that since the majority of cells have died, some of the observed proteomic/transcriptomic changes could actually be conducive towards halting aging rather than driving it? Put another way, in a sense, this method is selecting for long-lived cells and thus, some of the changes could reflect beneficial changes rather than deleterious ones, which is why those cells are still alive after so many replications. For example, transcriptional inhibition of biogenesis genes is a major aspect of the general stress response (which has been observed in Figure 3) that promotes longevity. Is it possible that these long-lived cells are a subpopulation with very high stress response levels and the method is biased toward these responses?

3) The authors should do a better job organizing and illustrating the rich dynamic data they have collected. For example, what is the fold-change time series (protein and transcript) of different gene groups (glycolysis, stress response, mitochondria)? Figure 3 did not provide information about actual fold-change values. Similarly, Figure 4 only shows 72 hr aged samples. It would be interesting to see how various gene groups alter their locations in the quadrant plots over time. In addition, the extraction of the detailed data from the supplemental tables is overly difficult. This point is especially important given the utility and importance of the dynamic data to the community. The authors need to facilitate access and plotting of the data (like the plots in Figure 2) in a straightforward manner. We would strongly suggest a dedicated website.

Technical concerns:

Since the authors presented a novel mother enrichment method, a few control experiments might be needed to further validate the approach:

1) The preparation for cell loading takes about 90 minutes and may involve a number of stress responses, including starvation in PBS and low temperature in 4 C water. Would the whole loading and column culturing processes affect the fitness and lifespan of cells? In accordance with this concern, the authors actually observed the induction of a number of general stress genes even at the early phase of lifespan (Figure 3). To prove that the cells are aging normally, a lifespan curve of cells growing under this column condition and a comparison with the lifespan curve of cells growing on regular culture plates (microdissection) or microfluidics (as in [38]) is sufficient.

2) It would be more convincing if the authors were able to use conventional methods to confirm the protein expression and transcriptional changes of a couple of the most interesting genes they identified (e.g. biogenesis genes?). For example, they could use biotin-affinity sorting or a genetic mother enrichment program to collect old mother cells, and use q-PCR and westerns to measure the transcription and expression levels.

3) Single mother cells divide with different rates and therefore at later time points the method collects mother cells with mixed ages. As shown in Figure 1—figure supplement 2, mother cells have a very wide distribution of ages at 44 hr and 68 hr. Does this complicate data interpretation? Is it possible to further separate mother cells into different age groups? At the very least a thorough and convincing discussion is needed.

---

## [Author Response]

Reviewer 1:

*1) The authors contend that they have isolated late age mothers. In Figure 1 they show a progressive decline in viability. How do they know that cells didn't die simply because they stayed longer in the column? They rely on additional criteria, but that data appears less convincing. Staining for bud scars provides the replicative age, and they track that with WGA intensity (Figure 1). However, the difference is ~ 2-fold, from start to finish. Why doesn't the signal increase more? One would expect that by 72h most cells would have >20 scars, while close to the start only ~2-3 scars. Likewise, cell size increases {less than or equal to} 2-fold (Figure 1). Others reported that the size increase is ~ 4-fold (DOI: 10.4161/cc.10.1.14455). Addressing these issues is important, for their thesis that properly aged cells have been isolated.*

The reviewer notes the flow cytometry measurements presented in Figure 1 (WGA staining for bud scars) and D (forward scatter) show a lower fold change with age than expected for bud scar increase and cell size increase, which makes the reviewer question if the cells in our experiment were properly aging. Indeed, in the way we had annotated and described the data of the figure, it was incomplete and could thus be wrongly understood (see following explanation). We thank the reviewer for pointing this out. We have fixed this in the revised manuscript.

The forward scatter measurements are only a very rough proxy for cell size, and the measurements of WGA intensity (which should be indicative for bud scars) were not corrected for the background fluorescence. Thus, true fold changes cannot be retrieved from these measurements. Furthermore, the reported WGA intensity values are just median WGA intensities of a mixed cell samples, rather than of separate mother and daughter populations (reason: overlapping spectral properties of the fluorophores for WGA (bud scar detection) and for Avidin (mother cell detection) made simultaneous use impossible). Thus, in addition, the reported WGA intensity is an underestimate of the true fluorescence intensity of mother cells. This was indeed not clearly indicated. We clarified these points in the revised version in the main text and legend to Figure 1. In addition, we added a confocal image of bud scars for readers to see bud-scar covered, late-age cells coming from the columns (which we could easily obtain from the column due to their enrichment).

Importantly, while as indicated above the WGA intensity and cell size proxy are qualitative assessments of aging in the column, the best indication that cells on the columns age properly is a comparison of the lifespan curves obtained from cells on our columns in comparison with classical lifespan data. To generate a high-quality lifespan curve from the cells as aged on our columns, for this revised version, we analyzed more data than we did previously (i.e. we used data from all mother cells in the column ( = from mixes 1, 2 and 3, and weighted by number of cells in each mix) rather than only data from the mother cells in mix 2 as we did previously). When comparing the resulting column-based lifespan curve with the lifespan curve generated with the microfluidic device (using the same strain and medium; 10.1073/pnas.1410024111) we found an excellent agreement (new Figure 1), which demonstrates proper aging of the cells on our columns.

*2) They report a strong un-coordination between the transcriptome and the proteome, getting worse in later stages. They obtained data for 1494 proteins and 4904 mRNAs. Hence, they did not sample >1/2 of the proteome. Still, sampling 1494 gene products represents a substantial dataset. My concerns have to do with their transcriptome/proteome comparisons. It has been argued that log2 comparisons (which is what they are doing) maybe prone to artifacts (PNAS, 107: 21487). A comprehensive comparison of available proteome, transcriptome and ribosome profiling data in yeast found widespread errors due to noisy data and incomplete coverage (DOI: 10.1371/journal.pgen.1005206), accounting for most of the discrepancies between mRNA and protein abundance. The data here are internally controlled, since the same methods are applied throughout the time series. Nonetheless, since proteome/transcriptome discrepancies are central, it will be prudent to analyze their data with different statistical models, and report on the overlap. They should also report how their findings for the early time points (which should be very similar to populations sampled previously) stack up against the literature.*

This is a good point. To address this point, we have performed further correlation analyses between the proteome and the transcriptome using multiple statistical tests, both on log2 and non-log2 transformed data (Pearson, Pearson with log2 transformation, Spearman, and Kendall). The results are presented in the new Figure 5—figure supplement 1. Here, regardless of the statistical test used, we obtained the same results, namely a reduction of correlation between the proteome and transcriptome.

In addition, because the study mentioned by the reviewer (Doi:10.1371/journal.pgen.1005206) has pointed out that a Spearman correlation may be more robust against noise and outliers in dataset comparisons, we have used the Spearman correlation in Figure 5, and for consistency, also for all other comparisons in the main figures. (Note, none of our original conclusions have changed).

As for the early time point comparison with the literature: we have noted in the main text that our correlation of the proteome and transcriptome is similar to what other studies have found in yeast: "To compare the changes between the proteome and transcriptome, we determined the non-parametric Spearman rank correlation, and found a starting correlation of 0.75, a value in agreement with other single-study comparisons between yeast proteomes and transcriptomes (7)."

*3) They conclude that changes in the translation machinery are causal to the aging process. This may be true, and fits well with the literature. However, as I understand it, they exclusively rely on the directionality of the network they have built (shown in Figure 5). For that, they look at whether the profile of a given gene product could predict the profile of another gene product. The ones that could* not *be predicted were clustered as causal. Hence, the approach appears to rely on* negative *evidence (lack of predictability) to arrive at such a powerful conclusion. For such a conclusion, I think some additional kind of evidence must be provided. At a minimum, they should look at the numerous mutations (mostly loss-of-function, but some gain-of-function) that have been reported to extend longevity in yeast. What is the enrichment for these gene products in their various clusters? Also, can they point to some new, unidentified ones from their study and test directly their role in determining replicative lifespan?*

This is also a good point. We realize that we did not mention an important aspect of our analysis. Note that the transcripts of the most causal cluster need to fulfill two criteria: these transcripts need to be less predicted by other transcripts (the negative evidence the reviewer is referring to), but in parallel they are required to have predictive ability on other transcripts (this is therefore a ‘positive’ requirement). Transcripts that do not have predictive ability, and are also themselves not predicted by others are removed from the analysis, when we move from the undirected network (Figure 6—figure supplement 1) to the directed network (Figure 6—figure supplement 1). In the revised manuscript we have explained better how the network is built and have clarified this aspect.

Due to the concern that the reviewer had with our analysis, the reviewer suggested we look at mutants that have published effects on lifespan and to compare their position in our transcriptome network. We hope that with clarifying the aspect above, such ‘validation’ with longevity mutants is no longer necessary. In fact, note that such a validation would rest on the assumption that genes that affect lifespan should be in more causal positions in the network. Unfortunately this is not necessarily true – i.e. a gene that has a product in a protein complex that losses stoichiometry with age may be downstream in the causal chain – but may nonetheless be vital for lifespan regulation.

Reviewer 2:

*1) The authors found that the biogenesis genes are enriched in the gene cluster, the transcriptional changes of which are "most causal" (ranked number 1) to other transcriptional changes during aging. They then immediately claim in the subsection “Newtwork interference identifies protein biogenesis related genes as casual in yeast aging” that "Taken together, as the biogenesis transcripts were the most predictive of all the changes we measured, our analyses support a model whereby the uncoupling of protein levels of biogenesis-related genes from their transcript levels is causal for the changes occurring in aging yeast". We think the authors need to better explain their reasoning here, because it is a key claim in the paper. Specifically, we would like to understand the following: (a) Is the decoupling of protein levels from transcript levels of biogenesis genes the causal factor for the transcriptiome changes, or the changes of both transcriptome and proteome, or their global decoupling during aging? (b) What is the mechanism for the decoupling to cause transcriptome changes, since it appears that the authors came up with this claim from their transcriptional network analysis? (c) If the authors want to claim that the decoupling of protein levels from transcript levels of biogenesis genes can cause the global decoupling of the proteome and transcriptome during aging (with which we are totally fine), then the best evidence would seem to arise from a similar network analysis for the decoupled time series of the decoupled genes (Q2 and Q4 in Figure 4). The network correlation analysis of the transcriptome in Figure 5 seems less than ideal in supporting this claim. To summarize this point, the authors present very rich and impressive data, but we believe that they need to be more specific about what they mean in order to draw such strong conclusions. A titrated summary would be sufficiently interesting: The decoupling of biogenesis protein levels from their transcript levels through some early-life molecular regulation (e.g. a general stress response?) can drive the global decoupling of the proteome and transcriptome, which is a consequence of aging and an important molecular phenotype of aged cells. Here, the overabundance of biogenesis genes is an early event driving a specific molecular consequence of aging, rather than the cause of all the transcriptome and proteome changes during aging.*

We agree that causality is a strong term. Indeed, inferring causality from omics data is a grand challenge. We agree that in the prior version of the manuscript we made at a too early stage statements that rather belong to a discussion. Accordingly, we have toned down the bold statements in the Results section, and instead we have expanded the Discussion. Here, we elaborate on the reviewer’s question, showing how it may be the case that the decoupling of protein levels from transcript levels of biogenesis genes (specifically) is causal in aging. We now make clear that we propose a model that is up for discussion: The model suggests that translation machinery can accumulate in young age (specifically decoupling the translation machinery’s proteins from transcripts, seen in Figure 5), further contributing to decoupling of the proteome from the transcriptome as a whole (since the translation machinery accounts for protein abundances, seen in Figure 5), and finally causing transcriptional responses in the cell (seen as responsive clusters in Figure 6, and protein complex deregulation in Figure 7). Furthermore, we acknowledge the possibility that causes upstream of the translation machinery may still be contributing to aging, which is also covered in the expanded Discussion. We hope that with these adjustments we have adequately addressed this comment of the reviewer.

2) It is stated that only 30% of the cells are alive at the final (12th) time point and the rest have died due to aging. Is it possible that since the majority of cells have died, some of the observed proteomic/transcriptomic changes could actually be conducive towards halting aging rather than driving it? Put another way, in a sense, this method is selecting for long-lived cells and thus, some of the changes could reflect beneficial changes rather than deleterious ones, which is why those cells are still alive after so many replications. For example, transcriptional inhibition of biogenesis genes is a major aspect of the general stress response (which has been observed in Figure 3) that promotes longevity. Is it possible that these long-lived cells are a subpopulation with very high stress response levels and the method is biased toward these responses?

The reviewer suggests that the changes we observed to occur with aging may reflect an enriched subpopulation of longer-lived cells, and thus the proteome/transcriptome we measure would eventually not be reflecting the natural aging process of a single cell. If this were the case, this would indeed be an important concern. Notably, this is an inherent problem to all population-level yeast aging studies. The best evidence against it, which we have described in the revised manuscript, is the validation of the protein abundance changes we measured with independent experiments. Specifically, when we compared single cell data obtained with GFP-tagged strains analyzed in microfluidic chips (the stress reporter HSP104 and TEF1), we found similar protein expression profiles with both methods and no evidence for a long lived subpopulation with different Hsp104 or Tef1 profiles. Thus, we conclude that our dynamic population-averaging data is not caused by the presence of specific long-lived subpopulations. We alert the reader of this potential issue in the main text, which is supported by a new supplementary figure.

*3) The authors should do a better job organizing and illustrating the rich dynamic data they have collected. For example, what is the fold-change time series (protein and transcript) of different gene groups (glycolysis, stress response, mitochondria)? Figure 3 did not provide information about actual fold-change values.*

To meet this concern, we added new figures to show the actual fold change of the proteins (Figure 3—figure supplement 3) and transcripts (Figure 4—figure supplement 3) that contribute to the 2-fold enrichment presented for the proteome in Figure 3 and for the transcriptome in Figure 4—figure supplement 3. Additionally, to further illustrate all dynamic fold changes we moved a supplemental figure into the main figures, creating a new main figure, Figure 4. Note that this results in the renumbering of figures. Finally, to provide further illustration of the data, we have added two examples of a protein complex losing stoichiometry with age (Figure 7, the proteasome and the vacuolar proton-transporting V-type ATPase, V1 domain, respectively).

*Similarly, Figure 4 only shows 72 hr aged samples. It would be interesting to see how various gene groups alter their locations in the quadrant plots over time.*

To address this concern, we have made a new figure (Figure 5—figure supplement 2) showing the ‘co-expression map’ at various time points in the aging process, highlighting subgroups of genes that contribute to the enrichment scores.

*In addition, the extraction of the detailed data from the supplemental tables is overly difficult. This point is especially important given the utility and importance of the dynamic data to the community. The authors need to facilitate access and plotting of the data (like the plots in Figure 2) in a straightforward manner.*

We have added individual supplemental tables of the final dataset, separate from the other tables used to generate the data. In this way, the final data of the aging proteome and aging transcriptome are easily accessible, and we have specified this in the text. These new tables are presented in [Supplementary-material SD4-data] and [Supplementary-material SD5-data] for the final proteomes and transcriptomes, respectively.

*We would strongly suggest a dedicated website.*

We have made contact with Dr. João Pedro de Magalhães, the curator/creator of various well known databases on the biology of aging (i.e. GenAge, AnAge, and the Digital Ageing Atlas), and are making arrangements to have an online, publically available, easily searchable database, making available our final proteome and transcriptome aging datasets in an integrated format. Such websites take time to develop, though we plan for this website to be operational by the end of the year.

*Technical concerns:*

*Since the authors presented a novel mother enrichment method, a few control experiments might be needed to further validate the approach: 1) The preparation for cell loading takes about 90 minutes and may involve a number of stress responses, including starvation in PBS and low temperature in 4 C water. Would the whole loading and column culturing processes affect the fitness and lifespan of cells? In accordance with this concern, the authors actually observed the induction of a number of general stress genes even at the early phase of lifespan (Figure 3). To prove that the cells are aging normally, a lifespan curve of cells growing under this column condition and a comparison with the lifespan curve of cells growing on regular culture plates (microdissection) or microfluidics (as in [41]) is sufficient.*

The comparisons of lifespan curves in the new Figure 1 demonstrate normal aging on the columns.

*2) It would be more convincing if the authors were able to use conventional methods to confirm the protein expression and transcriptional changes of a couple of the most interesting genes they identified (e.g. biogenesis genes?). For example, they could use biotin-affinity sorting or a genetic mother enrichment program to collect old mother cells, and use q-PCR and westerns to measure the transcription and expression levels.*

To address this point, we compared our data with information from the literature for single protein changes (Figure 3—figure supplement 4). Here, we have found good agreement. Specifically, we observed the protein levels of Hsp104, Tef1, and Vph1 levels to increase with age, consistent with what was previously described (references 1 and 2 below), protein levels of Nup116 and Nsp1 to decrease, as previously described (reference 3 below), Nup53 and Nup170 to be more stable (references 3 and 4 below) and levels of Tpo1 to increase and then to decrease, also as previously described (reference 4 below).

References:

1) Age related increase of Hsp104 and Tef1: doi:10.1371/journal.pone.0048275

2) Age related increase of Vph1: doi:10.1073/pnas.1113505109

3) Age related decline of Nsp1 and Nup116, while no significant changes were seen for Nup100 and Nup53: doi:10.1083/jcb.201412024

4) Age reated increase of Nup170: doi: 10.7554/eLife.03790.

5) Age related increase and decrease of Tpo1: doi:10.1038/ncb2085

6) Lifespan extension from Ras2 overexpression: http://www.jbc.org/content/269/28/18638.long

7) Lifespan extension from Mxr1 overexpression: doi:10.1073/pnas.0307929101

8) Lifespan extension from Vma1 overexpression: doi:10.1038/nature11654

Furthermore, we note that the main text already described a comparison with prior transcriptome studies (where aged cells were generated with conventional methods), with which we also found good agreement (provided in Figure 3). Therefore, we conclude that our datasets have been validated by both general and targeted studies looking at abundance changes in replicatively aging yeast.

*3) Single mother cells divide with different rates and therefore at later time points the method collects mother cells with mixed ages. As shown in Figure 1—figure supplement 2, mother cells have a very wide distribution of ages at 44 hr and 68 hr. Does this complicate data interpretation? Is it possible to further separate mother cells into different age groups? At the very least a thorough and convincing discussion is needed.*

The broad age distribution does not complicate the analysis but rather limits the resolution at higher ages. A further separation in age groups is not possible without extensive FACS sorting, which would most certainly affect the measured transcriptome and proteomes. We acknowledge the problem of broadening of the age distribution in the results section of the manuscript, but also now emphasize it in the Results section.